# The Anti-Acne Potential and Chemical Composition of Two Cultivated *Cotoneaster* Species

**DOI:** 10.3390/cells11030367

**Published:** 2022-01-21

**Authors:** Barbara Krzemińska, Michał P. Dybowski, Katarzyna Klimek, Rafał Typek, Małgorzata Miazga-Karska, Katarzyna Dos Santos Szewczyk

**Affiliations:** 1Department of Pharmaceutical Botany, Medical University of Lublin, 20-093 Lublin, Poland; barbara.krzeminska@umlub.pl; 2Department of Chromatography, Institute of Chemical Sciences, Faculty of Chemistry, Maria Curie Sklodowska University in Lublin, 20-031 Lublin, Poland; michal.dybowski@poczta.umcs.lublin.pl (M.P.D.); rafal.typek@poczta.umcs.lublin.pl (R.T.); 3Department of Biochemistry and Biotechnology, Medical University of Lublin, 20-093 Lublin, Poland; katarzyna.klimek@umlub.pl (K.K.); malgorzata.miazga-karska@umlub.pl (M.M.-K.)

**Keywords:** *Cotoneaster*, Rosaceae, antioxidant, anti-inflammatory, anti-acne, skin diseases, antimicrobial

## Abstract

In light of current knowledge on the role of reactive oxygen species and other oxidants in skin diseases, it is clear that oxidative stress facilitates inflammation and is an important factor involved in skin diseases, i.e., acne. Taking into consideration the fact that some *Cotoneaster* plants are valuable curatives in skin diseases in traditional Asian medicine, we assumed that thus far untested species *C. hsingshangensis* and *C. hissaricus* may be a source of substances used in skin diseases. The aim of this study was to evaluate the antioxidant, anti-inflammatory, antimicrobial, and cytotoxic activities of their various extracts. LC-MS analysis revealed the presence of 47 compounds (flavonoids, phenolic acids, coumarins, sphingolipids, carbohydrates), while GC-MS procedure allowed for the identification of 42 constituents (sugar derivatives, phytosterols, fatty acids, and their esters). The diethyl ether fraction of *C. hsingshangensis* (CHs-2) exhibited great ability to scavenge free radicals and good capacity to inhibit cyclooxygenase-1, cyclooxygenase-2, lipoxygenase, and hyaluronidase. Moreover, it had the most promising power against microaerobic Gram-positive strains, and importantly, it was non-toxic toward normal skin fibroblasts. Taking into account the value of the calculated therapeutic index (>10), it is worth noting that CHs-2 can be subjected to in vivo study and constitutes a promising anti-acne agent.

## 1. Introduction

The results obtained by Sarici et al. [1] concerning the evaluation of parameters associated with oxidative stress, such as nitric oxide (NO), xanthine oxidase (XO), malondialdehyde (MDA), superoxide dismutase (SOD), and catalase (CAT), in the venous blood of patients using spectrophotometrical methods indicate that oxidative damage of tissues is a significant etiological factor of acne. In view of the above, the role of oxidative stress in the etiology of acne seems to be unquestionable.

The cutaneous propionibacteria (*P. acnes*, *P. avidum*, *P. granulosum*, *P. propionicum*, and *P. lymphophilum*) are involved in the maintenance of healthy skin; however, they can also reveal adverse activity as opportunistic pathogens [2]. The three predominant genera, including *Propionibacteria*, *Staphylococci*, and *Corynebacteria*, form the microbial community established on the skin [3].

According to [4], the classification of *Propionibacterium* is extremely complicated. The major types of *Propionibacterium* (I, II, III clades) should be classified as *P. acnes* subsp. *acnes*, *P. acnes* subsp. *defendens*, and *P. acnes* subsp. *elongatum*, respectively. The latest research justifies the necessity to divide the genus *Propionibacterium* into four genera. Additionally, the cutaneous Propionibacterium, participating in seborrheic skin disorders, should be reclassified into the new genus *Cutibacterium*. Nevertheless, the previous classification is still valid and practiced among the medical microbiology community [4]. In our article, the new nomenclature (*C. acnes*) was used.

Although the association of acne lesion aggravation with the presence of *C. acnes* is well established, the mechanism of involvement of *C. acnes* in the formation of acne is not fully understood [5]. Nevertheless, it is well known that *C. acnes* participates in the production of pro-inflammatory cytokines, such as interleukin (IL)-8, IL-1β, IL-12, and tumor necrosis factor (TNF)-α, by stimulating keratinocytes, as well as phagocytic cells, which consequently leads to severe inflammation [5]. Furthermore, according to [6], enhancement of the mRNA expression levels of certain cytokines, such as (IFN)-γ, TNF-β, IL-8, IL-4, IL-10, IL-1α, IL-1β, and IL-17A, within acne lesions can be observed.

The presence of *C. acnes* on the skin leads to the formation of superoxide anions, which form peroxidants after combination with nitric oxide. As a consequence, this phenomenon causes keratinocyte damage [7]. 

Although synthetic antioxidant compounds are considered very effective, they also cause substantial side effects. Herbal sources possess the ability to participate in various phases of the oxidation mechanism. According to Soleymani and co-authors [8], phenolic compounds show significant anti-acne activity due to their ability to decrease oxidative stress. Many mechanisms of action are arranged in this biological activity and involve activation of TLR-2/4, p38/JNK/MAPK pathways, adjustment of MAPK signaling pathways, and regulation of Nrf2/Keapl-mediated antioxidant pathways. Additionally, upregulation of SOD, GP x, GSH, CAT, and heme oxygenase HO-1 was noted. Furthermore, polyphenolic compositions demonstrate the capability to downregulate endothelial ROS, NOX-4, and MDA (malondialdehyde) levels, which consequently leads to reduced amounts of H_2_O_2_ and MDA and to the inhibition of the expression of MAPKs, such as P38, ERK, and JNK. Pathways of particular significance, such as the ROS/MAPK/NF-ĸB pathway and PI3K/Akt/NF-ĸB pathway, are being suppressed. The decrease in lipid peroxidation linked with the downregulation of cytochrome P450 (CYP) expression and upregulation of STAT-1 2E1 expression seem to be notably relevant.

Medicinal plants have been traditionally used in the treatment of several human diseases, and their therapeutic properties have been assigned to different chemical compounds isolated from them. Of considerable importance, compounds with antioxidant properties may be found at great concentrations in plants and can be responsible for their preventive effects in various diseases caused by oxidative stress. Therefore, the antioxidant activities of plant extracts have prospective applications in healthcare [9].

A diversity of occurrence among the *Cotoneaster* genus is worth underlining and encompasses river valleys and banks, woods, thickets, rocky, and calcareous sites, as well as mountain areas approximately 800–4100 m above sea level. The individual species are indigenous, particularly to the Palearctic region (temperate Asia, Europe, and North Africa). Nevertheless, they are cultivated not only throughout Europe but also in various parts of the world due to their significant aesthetic qualities: white, red, or pink flowers and red, brownish red, orange, or black fruits. The pivotal area of occurrence, with 60% of species, can be observed in mountain regions of China [10]. *Cotoneaster hsingshangensis* is indigenous to China, and *C. hissaricus* is vernacular throughout Asia. These two species, similar to many others of *Cotoneaster*, are cultivated in Poland as ornamental plants in parks, gardens, and other urban places.

It has been reported that many *Cotoneaster* species have valuable biological properties and have been used in traditional medicine in many countries [11]. Swati and co-authors [12] reported that fruits of *Cotoneaster microphyllus* are applied on skin against irritation. Furthermore, leaves of *C. microphyllus* are used for dermatitis. The medicinal uses of *Cotoneaster* species in skin disorders have been reported in Pakistan, Turkey, India, Lebanon, and Iran as therapeutic agents in the treatment of cuts and wounds [13]. In addition, an anti-itching effect of *Cotoneaster* has been observed [14,15]. According to Akbar [16], *Cotoneaster* removes hyperpigmentation of the skin.

Taking into consideration the fact that some representatives of *Cotoneaster* are valuable curatives applied in various skin diseases in traditional Asian medicine, we assumed that thus far untested species—*C. hsingshangensis* and *C. hissaricus*—may be a source of active substances used in skin diseases. In view of the above, the present study reports primarily on a thorough examination of the composition (especially phenolic components) and biological activities with an emphasis on dermatological diseases, such as acne. An exploration of *Cotoneaster* species has been undertaken to provide an explanation for their capability to ameliorate skin conditions.

## 2. Materials and Methods

### 2.1. Chemicals and Reagents 

Ascorbic acid, 2,2-diphenyl-1-picrylhydrazyl radical (DPPH^•^), 2,2′-azino-bis-(3-ethyl-benzothiazole-6-sulfonic acid) (ABTS^•+^), (−)-epigallocatechin gallate (EGCG), nordihydroguaiaretic acid (NDGA), indomethacin, hyaluronidase from bovine tests, hyaluronic acid sodium salt from rooster comb, Folin–Ciocalteu reagent, ethylene-diaminetetraacetic acid, disodium dihydrate (Na_2_EDTA*2H_2_O), Tricine (≥99%; titration) were obtained from Sigma-Aldrich (Steinheim, Germany). Phosphate-buffered saline (PBS) was purchased from Gibco (Carlsbad, CA, USA). Reference substances were supplied by ChromaDex (Irvine, CA, USA), while acetonitrile, formic acid, and water were supplied for LC analysis by Merck (Darmstadt, Germany). All other chemicals were of analytical grade and were obtained from the Polish Chemical Reagent Company (POCH, Gliwice, Poland).

### 2.2. Plant Material 

Leaves of *Cotoneaster hsingshangensis* Fryer et B.Hylmö (Figure 1), as well as *C. hissaricus* Pojark. (Figure 2) were collected in the Maria Curie-Skłodowska University (UMCS) Botanical Garden in Lublin (Poland), at an altitude of 181.2 m a.s.l. (coordinates 52°14′34.4″ N 17°05′32.4″ E and 46°57′51.7″ N 142°45′22.1″ E, respectively) in September 2020. Taxonomical identification was confirmed by Dr. A. Cwener, an employee of the Botanical Garden who specializes in *Cotoneaster*. *C. hsingshangensis* was introduced to cultivation in the Botanical Garden in Lublin in 1966. This species was derived from the Institute of Dendrology of the Polish Academy of Sciences, Kórnik Arboretum. *C. hissaricus* was introduced to the Botanical Garden in Lublin in 2003 from Sakhalin Botanical Garden. Voucher specimens were deposited in the Department of Pharmaceutical Botany (CHs-0921 and CHi-0921). 

### 2.3. Preparation of the Extracts

The plant materials were dried in the shade at 24 °C (±0.5 °C) to achieve a constant weight [17]. Extracts were prepared using a mixture of methanol–acetone–water (3:1:1, *v*/*v*/*v*; 3 × 100 mL), and then sonicated at a controlled temperature (40 ± 2 °C) for 30 min [13]. The combined extracts were filtered, concentrated under reduced pressure, and after freezing, lyophilized in a vacuum concentrator (Free Zone 1 apparatus; Labconco, Kansas City, KS, USA) to obtain dried residues. The obtained extracts were dissolved in hot water, filtered after 24 h, and subjected to liquid-liquid extraction with diethyl ether, ethyl acetate, and n-butanol, successively. The obtained fractions of diethyl ether, ethyl acetate, and n-butanol, as well as the water residue, were evaporated in vacuo and lyophilized using a vacuum concentrator. 

### 2.4. Total Flavonoid, Phenolic, and Phenolic Acids Content

Total flavonoid (TFC) and total phenolic content (TPC) were established using colorimetric assays as described previously [18]. The absorbance was measured at 430 and 680 nm, respectively, using a Pro 200F Elisa Reader (Tecan Group Ltd., Männedorf, Switzerland). TPC was estimated from the calibration curve (R^2^ = 0.9845), using gallic acid as a standard (concentration ranged 0.002–0.1 mg/mL). The results were expressed as mg of gallic acid equivalent (GAE) per 1 g of dry extract (DE). TFC was estimated from the calibrated curve (R^2^ = 0.995), using quercetin (0.004–0.11 mg/mL) as a standard. The results were expressed as mg of quercetin equivalent (QE) per 1 g of DE. Total phenolic acid (TPAC) content was assessed using Arnov’s reagent as described in the Polish Pharmacopoeia IX (an official translation of PhEur 7.0) [17]. The absorbance was measured at 490 nm. TPAC was estimated from the calibration curve (R^2^ = 0.9999), using caffeic acid as a standard in a concentration of 3.36–23.52 μg/mL. The results were ex-pressed as mg of caffeic acid equivalent (CAE) per 1 g of DE.

### 2.5. LC-MS Analysis

The chromatographic measurements were performed using the LC/MS system from Thermo Scientific (Q-EXATCTIVE and ULTIMATE 3000, San Jose, CA, USA) equipped with an ESI source. The ESI was operated in negative polarity modes under the following conditions: spray voltage—3.5 kV; sheath gas—40 arb. units; auxiliary gas—10 arb. units; sweep gas—10 arb. units; and capillary temperature—320 °C. Nitrogen (>99.98%) was employed as sheath, auxiliary, and sweep gas. The scan cycle used a full-scan event at a resolution of 60,000. A Gemini C18 column (4.6 × 100 mm, 3 μm) (Phenomenex, Torrance, CA, USA) was employed for chromatographic separation, which was performed using gradient elution. Mobile phase A was 25 mM formic acid in water; mobile phase B was 25 mM formic acid in acetonitrile. The gradient program started at 5% B, increasing to 95% for 60 min, followed by isocratic elution (95% B) for 10 min. The total run time was 70 min at the mobile phase flow rate 0.4 mL/min. The column temperature was 25 °C. In the course of each run, MS spectra in the range of 100–700 *m*/*z* were collected continuously. Additionally, the MS2 functions were used to carry out a detailed qualitative analysis. The collision energy for each examined compound was 25%.

The amounts of the identified compounds were carried out based on the calibration curves obtained for the standard. In the case of quantitative analysis of compounds that do not have standards, calibration curves for substances of similar structure were used. All the results are presented as the mean of three independent measurements (*n* = 3).

### 2.6. GC-MS Analysis

The qualification of the sample extract was performed using a GC-MS/MS system (GCMS-TQ8040; Shimadzu, Kyoto, Japan) equipped with a ZB5-MSi fused-silica capillary column (30 m × 0.25 mm i.d., 0.25 μm film thickness; Phenomenex, Torrance, CA, USA). Grade 5.0 helium was used as the carrier gas. Column flow was 1 mL/min. The injection of a 1 μL sample was performed using an AOC-20i + s type autosampler (Shimadzu, Kyoto, Japan). The injector was working at a temperature of 310 °C. The following temperature program was applied: the oven temperature was held at 60 °C for 2 min and was subsequently increased linearly at a rate of 6 °C/min to 310 °C, where it was held for 15 min. The mass spectrometer was operated in EI mode at 70 eV; the ion source temperature was 225 °C. The mass spectra were measured in the range 40–450 amu. The amounts of the individual analytes were estimated by the peak normalization method.

### 2.7. Antioxidant Activity 

All antioxidant and enzyme inhibitory assays were done in 96-well plates (Nun-clon, Nunc, Roskilde, Denmark) using Infinite Pro 200F Elisa Reader (Tecan Group Ltd., Männedorf, Switzerland). The experiments were performed in triplicate.

#### 2.7.1. DPPH^•^ Assay

The 2,2-diphenyl-1-picryl-hydrazyl (DPPH^•^) free radical scavenging activity of *Cotoneaster* extracts and the positive control—ascorbic acid (AA)—was studied using the method described previously [18], but with some modifications. After 30 min of incubation at 28 °C, the decrease in DPPH^•^ absorbance, caused by the tested extracts, was measured at 517 nm. The results were expressed as values of IC_50_.

#### 2.7.2. ABTS^•+^ Assay

The ABTS^•+^ decolorization assay was the second method applied for the assessment of antioxidant activity [18]. The absorbance was measured at 734 nm. Trolox was used as a positive control. The results were expressed as values of IC_50_.

#### 2.7.3. Metal Chelating Activity (CHEL)

The metal chelating activity was established using the method described by Guo et al. [19], modified in our previous study [18,20]. The absorbance was measured at 562 nm. As a positive control, Na_2_EDTA*2H_2_O was used. Results were expressed as the IC_50_ values of the *Cotoneaster* extracts based on concentration-inhibition curves.

### 2.8. Enzyme Inhibitory Activity

#### 2.8.1. Cyclooxygenase-1 (COX-1) and Cyclooxygenase-2 (COX-2) Inhibitory Activity

The extracts of *Cotoneaster* species were examined for cyclooxygenase-1 (COX-1) and cyclooxygenase-2 (COX-2) inhibitory activity using a COX (ovine/human) Inhibitor Screening Assay Kit (Cayman Chemical, MI, USA) according to the protocol of the manufacturer. The extracts were tested at different concentrations. Indomethacin was used as a positive control.

#### 2.8.2. Lipoxygenase Inhibitory Activity

Anti-lipoxygenase activity of *Cotoneaster* extracts was determined using the Lipoxygenase Inhibitor Screening Assay Kit (Cayman Chemical, MI, USA) according to the protocol of the manufacturer. The extracts were tested at different concentrations. The effective concentration (μg/mL) in which lipoxygenase activity is inhibited by 50% (IC_50_) was estimated graphically. Nordihydroguaiaretic acid (NDGA) was used as a positive control. 

#### 2.8.3. Hyaluronidase Inhibitory Activity

Anti-hyaluronidase activity was established using the method described by Liyanaarachchi et al. [21]. After 20 min incubation at 37 °C, the absorbance was measured at 585 nm. The extracts were tested at different concentrations. Epigallocatechin gallate was used as a positive control.

### 2.9. Bacterial Strains

The antibacterial power of *Cotoneaster* extracts was determined using bacterial strains causing skin diseases (including seborrhea and acne). We used microaerobic Gram-positive bacteria: *Cutibacterium granulosum* PCM 2462, *C. acnes* PCM 2334, *C. acnes* PCM 2400, (possessed from the Polish Collection of Microorganisms PCM Institute of Immunology and Experimental Therapy Polish Academy of Sciences, Poland, as *Propionibacterium*, now—*Cutibacterium*), *C. acnes* ATCC 11827; aerobic Gram-positive: *Staphylococcus epidermidis* ATCC 12228 and *S. aureus* ATCC 25923; and aerobic Gram-negative strains *Pseudomonas aeruginosa* ATCC 27853 and *Escherichia coli* ATCC 25992. Each bacterial strain was pre-incubated overnight at 37 °C on agar plates. Mueller–Hinton (BioMaxima S.A., Lublin, Poland) agar or broth (MH-agar, MH-broth) for aerobic strains and Brain–Heart Infusion (Oxoid Ltd, Altrincham, England) agar or broth (BHI-agar, BHI-broth) for microaerobic bacteria were used. The bacteria were suspended in 5 mL of sterile saline water, and the absorbance of this inoculum was adjusted to 108 CFU/mL (0.5 Mc’Farland scale). 

### 2.10. Disc Diffusion Assay 

This disc diffusion assay can evaluate the antibacterial potency of tested extracts and was made according to proven methods [22,23]. Briefly, solid medium in Petri dishes was smeared with inoculum of 0.5 Mc’Farland. The plant extracts were dissolved in DMSO (10 mg/mL) then loaded over sterile filter paper discs (8 mm in diameter) to obtain a final concentration of 0.1 mg per disc. Inoculated plates with samples were incubated at 37 °C for 24 h (aerobic stains) or 48 h (microaerobic bacteria).

The zones of growth inhibition around plant samples were measured [mm] and recognized as antibacterial potency. The larger the zone of growth inhibition, the greater the antibiotic activity. 

### 2.11. Minimum Inhibitory Concentration (MIC) and Minimum Bactericidal Concentration (MBC) Determination 

The MIC is the lowest concentration of a test compound that inhibits microbial proliferation. The test examined the MIC of the *Cotoneaster* extracts against *Staphylococcus aureus* ATCC 25923, *S. epidermidis* ATCC 12228, *C. acnes* ATCC 11827, *C. acnes* PCM 2334, *C. acnes* PCM 2400, and *C. granulosum* PCM 2462, whose optical density was 0.5 Mc’Farland scale. Double microdilution in the 96-well plate assay was used according to the CLSI method with some modification, as described in our paper [23,24]. Such an amount of the extract solution to the wells was added, that using the principle of double microdilution to obtain final concentrations in the range of 3.9–1000 µg/mL. Then, each well was inoculated (2 µL of inoculum density of 0.5 Mc’Farland). A background control (broth with extract), a negative control (the broth alone), and a positive control (broth and inoculum) were also prepared. The plates were incubated under conditions proper to bacterial growth (aerobic bacteria at 37 °C, for 24 h; anaerobic bacteria at 37 °C, for 48 h). Finally, microbial growth was determined using a BioTek Synergy H4 (BioTek, Winooski, VT, USA) plate reader at a wavelength of 600 nm. 

MBC is the lowest concentration of the agent with bactericidal properties. To determine this, 96-well plates obtained in the previous experiment were used to determine the MIC of the tested extracts. Ten µL of solutions taken from the wells in which no bacterial growth was observed were applied to the new Petri plates. Then, the plates were incubated under optimal conditions for the growth of the given bacteria. After the incubation was completed, a visual assessment of the agar surface of the plates was made. The MBC was considered to be the one in which there was no visible bacterial growth on the solid medium.

To visualize the obtained results, it is advantageous to present the MBC/MIC ratio. Thus, MBC/MIC values ≤4 demonstrate that the agent is bactericidal. MBC/MIC values >4 indicate the bacteriostatic nature of the tested agents [25].

Microbiological tests were performed in three separate experiments (*n* = 3).

### 2.12. Cytotoxic Activity

This experiment was carried out according to the protocol described by us previously [26]. The cytotoxic activity of selected substances was evaluated toward the BJ cell line (normal human fibroblasts, ATCC CRL-2522TM). Briefly, the BJ cells were seeded in 96-well plates, and then after 24 h incubation at 37 °C in suitable conditions (5% CO_2_, 95% air), the culture medium was replaced with two-fold serial dilutions of investigated substances (1.95–1000 μg/mL). After 24 h incubation, the cell viability was assessed using the MTT assay. The obtained data were presented as mean values ± standard deviations (SD). These results were subjected to four-parameter nonlinear regression analyses (GraphPad Prism 5, version 5.04, GraphPad Software, San Diego, CA, USA) in order to determine values of half-maximum cytotoxic concentration (CC_50_). 

### 2.13. Statistical Analysis

The results were expressed as mean values ± standard deviation (SD) of three independent experiments. The data from cell culture experiments were subjected to statistical analysis using unpaired Student’s *t*-test or One-Way ANOVA test, followed by a Tukey’s multiple comparison test, and differences were considered significant when *p* < 0.05 (GraphPad Prism 5, version 5.04, GraphPad Software, San Diego, CA, USA), whereas Principal Component Analysis was carried out in R version 3.6.3 (64-bit, Windows 10), using built-in “prcomp” function.

## 3. Results

### 3.1. Phytochemical Analysis

Total phenolic content (TPC) for *Cotoneaster* extracts and fractions was determined using Folin-Ciocalteu reagent, and the results were estimated as gallic acid equivalents (GAE) per g of dry extract (DE) (Table 1). Our results showed that the leaves of *C. hissaricus* (CHi) have the highest phenolic content (296.13 ± 1.52 mg GAE/g DE) than *C. hsingshangensis* (CHs) (193.84 ± 1.14 mg GAE/g DE). The results obtained in our study were better compared to data presented for extracts derived from the leaves of other *Cotoneaster* species. For instance, Kicel et al. [27] demonstrated that TPC for the 70% aqueous methanolic extracts of leaves of different *Cotoneaster* species cultivated in Poland varied from 51.7 (*C. tomentosus*) to 154.3 mg GAE/g (*C. bullatus*) of dry weight of the plant material. The same authors [28] also found lower amounts of phenolic compounds in 70% aqueous methanolic extracts of the fruits (from 26 to 43.5 mg GAE/g PM) of different *Cotoneaster* species. 

In our study, we also examined the content of phenolics in fractions obtained after fractionating crude extracts between solvents of different polarity. The results showed that the TPC level was in the range of 83.87 ± 0.23 (water fraction of *C. hissaricus*) to 559.77 ± 3.76 mg GAE/g DE (ethyl acetate fraction of *C. hsingshangensis*). Kicel and co-authors [29] obtained higher values with the highest TPC level for ethyl acetate fractions (470.9–650.8 mg GAE/g dw) and diethyl-ether fractions (453.1–546.9 mg GAE/g dw) of defatted methanol extracts. 

Thus, the results obtained in our study indicated that the leaves of *C. hissaricus* and *C. hsingshangensis* are a rich source of phenolic compounds. 

The total flavonoid content of the leaves of *C. hissaricus* and *C. hsingshangensis* was estimated by a previously described colorimetric method [27]. The data were expressed as quercetin equivalents (QE) per g of dry extract (DE). The results presented in Table 1 show that the content of total flavonoids in both species is at an average level. The comparable content was noted for the leaves of *C. hissaricus* (CHi) and *C. hsingshangensis* (CHs) (25.38 ± 2.35 and 47.72 ± 0.37 mg QE/g DE, respectively). The results obtained in our study were higher than those found by Mahmutović-Dizdarević et al. [11]. In their study, quantitative estimation revealed that methanolic extracts from leaves of *C. tomentosus* possessed 18.17 ± 0.30 mg QE/g dw. flavonoid content, followed by *C. integerrimus*—16.42 ± 0.35 mg QE/g dw. and *C. horizontalis*—10.55 ± 0.51 mg QE/g dw. Moreover, they found that fruits of these three species contain a lower amount of flavonoids—2.76–9.38 mg QE/g dw. A higher amount of flavonoids was found in the methanolic extract of leaves of *C. wilsonii* Nakai (36.46 ± 1.89 mg QE/g dw), while in the stems and fruits, the content of flavonoids was lower (6.09 ± 0.71 and 0.23 ± 0.20 mg QE/g dw, respectively) [30]. Kicel et al. [29] also obtained lower results for defatted methanol extracts of the three *Cotoneaster* species cultivated in Poland, and the values were from 14.70 ± 0.05 to 67.71 ± 0.80 mg/g dw. These authors obtained slightly higher results for the fractions obtained from crude extracts compared to the results in our study. The highest level of flavonoids was determined for the ethyl acetate fraction of *C. integerrimus* leaves (403.56 ± 7.48 mg/g dw). In our study, the lowest amount of total flavonoids was found in water fractions of *C. hissaricus* and *C. hsingshangensis* (0.12 ± 0.05 and 0.23 ± 0.02 mg QE/g DE, respectively) and the highest was in the ethyl acetate fraction of *C. hsingshangensis* (252.27 ± 0.24 mg QE/g DE).

The total phenolic acid content (TPAC) in *Cotoneaster* extracts is presented in Table 1. As for the content of polyphenols and flavonoids, a higher content of phenolic acids was noted for crude extract (61.27 ± 0.93 mg CAE/g DE) and fractions (19.65 ± 0.30–91.95 ± 0.48 mg CAE/g DE) of *C. hsingshangensis*. 

The use of plants in the cure of different diseases depends on their phytochemical composition. Our research of the presence of active compounds in the crude methanol–acetone–water (3:1:1, *v/v/v*) extracts and different fractions (diethyl ether, ethyl acetate, butanol, and aqueous residual) indicated that methanol–acetone–water extracted the greatest range of constituents from the leaves of both *Cotoneaster* species. Therefore, in our study, we used only the crude extracts for the LC-MS and GC–MS analyses.

Thus, in the next step of our study, the chemical composition of the extracts obtained from *Cotoneaster* extracts was investigated using the LC-MS method. Table 2 shows 47 identified compounds, including their molecular formula, theoretical and experimental molecular mass, both errors in ppm and mDa, and the fragments. In our study, flavonoids, phenolic acids, coumarins, cyanogenic glycosides, sphingolipids, and carbohydrates were identified using LC-MS analysis. The chromatograms with marked main peaks are displayed in Figure 3 and Figure 4. The results of the quantitative analysis are presented in Table 3. The amounts of the identified compounds were carried out based on the calibration curves obtained for the standard. In the case of quantitative analysis of compounds that do not have standards, calibration curves for substances of similar structure were used. 

Among flavonoids, the most abundant in both species were quercetin derivatives. In the leaves of *C. hissaricus* rutin (18,028 ± 650 μg/g DE), isoquercitrin (10,079 ± 353 μg/g DE), hyperoside (9119 ± 331 μg/g DE), 5,7,2′,5′-tetrahydroxyflavanone 7-*O*-glucoside (8067 ± 290 μg/g DE), and quercetin 3-*O*-(2″-*O*-xylosyl)galactoside (7318 ± 289 μg/g DE) were found in the largest amount. In the leaves of *C. hsingshangensis*, isoquercitrin (8926 ± 327 μg/g DE), quercitrin (6726 ± 251 μg/g DE), and hyperoside (6184 ± 237 μg/g DE) were observed in the greatest amount. Isoquercitrin, rutin, hyperoside, and quercitrin were previously identified as dominant in the other *Cotoneaster* species [27,28,31,32,33,34,35,36,37]. Vitexin 2″-*O*-arabinoside, vitexin-2″-*O*-rhamnoside, and 5-methylgenistein, which were found in large quantities in both studied species, were previously noticed only in leaves of *C. thymaefolia* [32], aqueous alcoholic extracts of leafy twigs of *C. obricularis* [38], and chloroform extracts of *C. simonsii* leafy twigs [39], respectively. Subsequent compounds of rare occurrence in the *Cotoneaster* genus are 5,7,2′,5′-tetrahydroxyflavanone and its 7-*O*-glucoside, which were identified in *C. thymaefolia* leaves [32]. It is worth noting that rare in nature flavonoids, such as biochanin A 7-*O*-glucoside (sissotrin) and 5-methylgenistein-4′-*O*-glucoside were observed in both studied species. Previously, sissotrin was found only in the n-butanol fraction of the methanolic extract of leaves of *C. mongolica* [37], methanol extract of *C. serotina* flowers [40], and methanol extract of flowers and fruits of *C. pannosa* [40]. 5-methylgenistein-4′-*O*-glucoside was previously observed only in the chloroform extract of leafy twigs of *C. simonsii* [39]. Quercetin 3-*O*-gentiobioside (quercetin 3-*O*-β-d-glucopyranosyl(l-6)glucopyranoside) is noteworthy regarding the fact that it was one of the first compounds isolated from the *Cotoneaster* genus and is infrequent, being only reported in *C. oligantha* stems [41]. Furthermore, kaempferol 3-*O*-glucoside (astragalin) is of rare occurrence in the *Cotoneaster* genus and reported only in the methanol extract of *C. mongolica* leaves by Odontuya and co-authors [37].

The second largest group of active compounds were phenolic acids, and among them, chlorogenic acid, gentisic acid 2-*O*-glucoside, and caffeoylmalic acid were the most abundant in both species. Apart from these typical phenolic acids, cotonoate A and horizontoate A were observed in the leaves of *C. hissaricus* and *C. hsingshangensis*, but only in *C. hissaricus* was cotonoate A identified in quantifiable amounts (1564 ± 55 μg/g DE). In previous research, cotonoate A and horizontoate A were isolated only from the leafy twigs of the *C. racemiflora* chloroform soluble fraction of the methanolic extract [14] and methanolic extract of *C. horizontalis* [15], respectively.

Chlorogenic acid seems to be a ubiquitous ingredient among the *Cotoneaster* genus. What most attracts attention is that both studied extracts of leaves of *C. hissaricus* and *C. hsingshangensis* constitute a significant percentage of this acid (37,932 ± 1330 and 60,043 ± 2231 μg/g DE, respectively). According to [42], this compound participates in diminishing *P. acnes*-induced matrix metalloproteinase-9 levels, obstructing nuclear factor-κB (NF-κB) activation, inactivating mitogen-activated protein kinases (MAPK), and decreasing the migration of neutrophils and interleukin (IL)-1β+ populations in vivo. Therefore, it is increasingly being recognized as possibly efficacious in the management of dermatological conditions, such as acne.

Among the other polyphenolic compounds, some coumarins were identified in the leaves of *C. hissaricus* and *C. hsingshangensis,* and the most abundant in both species was scopoletin (12,219 ± 440 and 10,481 ± 371 μg/g DE, respectively). This compound was isolated earlier from the leafy twigs of *C. racemiflora* from methanolic extract [43] and the ethyl acetate-soluble fraction of methanolic extract [44,45].

In both studied species, great amounts of mannitol were also observed (*C. hissaricus*—6834 ± 249 μg/g DE and 3104 ± 123 μg/g DE—*C. hsingshangensis*). This compound is the predominant polysaccharide of manna, which is produced by the young shoots of *C. discolor*, *C. nummularius*, *C. tricolor*, and *C. nummularioides* [46]. From cyanogenic glycosides, prunasin, and amygdalin were found. These glycosides were also previously identified in the fruits and leaves of *C. congesta*, *C. praecox*, and *C. integerrimus* [47] and in the ethanol extract of *C. horizontalis* leafy twigs [48].

Between phenolic compounds occurring in *C. hsingshangensis* and *C. hissaricus*, caffeic acid [44,45], ferulic acid [49,50], chlorogenic acid [51], cinnamic acid [52], quercetin [53], and vitexin [54] are substances with proven anti-acne properties on skin.

The appropriate GC-MS procedure allowed for the identification of 42 and 41 compounds in the leaves of *C. hissaricus* (Table 4, Figure 5) and *C. hsingshangensis* (Table 5, Figure 6), respectively. Among the identified compounds, four main groups of analytes can be distinguished—sugar derivatives, phytosterols, fatty acids, and their esters. 

The dominant group of compounds included in the extract from *C. hissaricus* are sugar derivatives; their content exceeds 37%. 1,3:2,5-dimethylene-l-rhamnitol, the content of which in the extract is 12.29%, and 2-deoxy-d-galactose, the content of which is 14.62%, are noteworthy. In the case of the extract of *C. hsingshangensis*, the content of sugar derivatives does not exceed 30%. The estimated amount of 1,3:2,5-dimethylene-l-rhamnitol (14.62%) is at a similar level of concentrations, while the content of 2-deoxy-d-galactose is nearly three times lower. The second major group of identified compounds is phytosterols. The content of those compounds in the extract of *C. hissaricus* exceeds 21%, while the amount of the main phytosterol, phytol acetate, is 15.79%. The extract of *C. hsingshangensis*, in turn, contains an approximately 25% higher concentration of phytosterols (more than 26%). The third major group of compounds in both extracts was fatty acids and their esters. Among the most important are hexa-decanoic acid, methyl ester (which constitutes 10.07%—CHi; 17.02%—CHs), followed by palmitic acid (6.09%—CHi; 5.14%—CHs), and linolenic acid (2.14%—CHi; 3.39%—CHs). Linolenic and palmitic acids were previously recorded as the major fatty acids in the fruits of *C. zabelii*, *C. splendens*, *C. hjelmqvistii*, and *C. horizontalis* [28] and the aerial parts of *C. horizontalis* [55].

Based on the GC/MS analysis, it was found that most of the identified compounds were present in both *C. hissaricus* and *C. hsingshangensis*. However, ((E)-2,6-dimethoxy-4-(prop-1-en-1-yl)phenol, 1-docosanol, methyl ether, 3-hydroxy-β-damascone, cholesta-4,6-dien-3-ol, (3β)-, octatriacontyl trifluoroacetate, and urs-12-en-28-al were only detected in *C. hissaricus* leaves, and 2,6,10,14,18-pentamethyl-2,6,10,14,18-eicosapentaene, benzenesulfonamide, N-butyl-, d-glucitol, 2,5-anhydro-1-*O*-octyl-, lauric acid methyl ester, and tridecanoic acid were only found in *C. hsingshangensis*.

It Is worth mentioning that the pronounced bioactivity of phenols toward skin disorders is enhanced by the simultaneous presence of such components as 6-hydroxy-4,4,7a-trimethyl-5,6,7,7a tetrahydrobenzofuran-2(4H)-one, which was reported as a strong anti-inflammatory agent [56], and which was identified in both studied *Cotoneaster* species.

A broad variety of active compounds in *Cotoneaster* species lead to overlapping activities of miscellaneous agents, hence acting on several levels of different diseases caused by oxidative stress is possible. 

### 3.2. Biological Activity

Taking into account that both oxidation and inflammation play significant roles in disease pathogenesis, recent research has focused on the evaluation of the antioxidant and anti-inflammatory activity of plants. In our study, we examined in vitro antioxidant, anti-cyclooxygenase, anti-lipoxygenase, anti-hyaluronidase, antibacterial, and cytotoxic activities of leaves of *C. hissaricus* and *C. hsingshangensis*.

#### 3.2.1. Antioxidant Activity

The antioxidant activity was studied on the microplate scale in cell-free systems. The *Cotoneaster* extracts were evaluated in a concentration ranging from 10 to 150 μg/mL. It was demonstrated that all investigated extracts exhibited moderate scavenging capacity in a concentration-dependent manner (Table 6). For comparison, the radical scavenging activity of ascorbic acid (AA; IC_50_ = 4.75 ± 0.16 μg/mL), quercetin (IC_50_ = 2.05 ± 0.10 μg/mL) and Trolox (IC_50_ = 3.68 ± 0.09 μg/mL) were tested in the same conditions. The highest DPPH scavenging activity was shown for the ethyl acetate fraction (CHs-4) (IC_50_ = 2.08 ± 0.03 μg/mL), followed by the butanol fraction (CHs-3), and the diethyl ether fraction (CHs-2) of *C. hsingshangensis* (IC_50_ = 3.43 ± 0.02 and 4.15 ± 0.05 μg/mL, respectively). The weakest activity was noted for the water fraction (CHi-1; IC_50_ = 32.37 ± 0.19 μg/mL) and crude extract of *C. hissaricus* (CHi; IC_50_ = 21.73 ± 0.13 μg/mL).

The results of other published reports seem to be difficult to compare due to the other conditions used during experiments. Nevertheless, antioxidant activity using the DDPH assay was studied for different fractions of 70% methanol extract of the leaves of three *Cotoneaster* species cultivated in Poland [29]. The authors found that the studied extracts and fractions possessed significant scavenging effects with IC_50_ values ranging from 3.19 μg/mL for the ethyl acetate fraction of *C. bullatus* to 27.88 μg/mL for the water residue of *C. integerrimus* (the IC_50_ values for positive controls—quercetin, (−)-epicatechin, chlorogenic acid, butylated hydroxyanisole, and Trolox were 1.70, 2.35, 4.60, 2.90, and 4.05 μg/mL, respectively). Holzer et al. [26] also studied antioxidant capacity with DPPH tests of different extracts of sterile shoots, old stems, and leaves of *C. melanocarpus*. They found that the water extract had IC_50_ values from 53.53 to 86.72 μg/mL, methanolic extracts—30.91–106.41 μg/mL, ethyl acetate extracts—74.88–200 μg/mL, dichloromethane extracts—134.60–200 μg/mL. Ten polyphenol compounds isolated from the leaves of *C. bullatus* and *C. zabelii* and crude hydroalcoholic extracts were also evaluated by the DPPH radical scavenging assay. All these compounds showed significant inhibitory activity. Importantly, (−)-epicatchin (IC_50_ = 2.35 μg/mL), procyanidin B2 (IC_50_ = 2.46 μg/mL), and procyanidin C1 (IC_50_ = 2.55 μg/mL) showed stronger activity compared to Trolox (IC_50_ = 4.06 ± 0.11 μg/mL), which was used as a positive control [36]. Significant antioxidant activities were also observed for the compounds isolated from the leafy twigs of *C. racemiflora*, including racemiside (IC_50_ = 11.1 μM), scopoletin (IC_50_ = 3.8 μM), 7,8-dimethoxy-6-hydroxycoumarin (IC_50_ = 1.8 μM), 3,3′,4′-tri-*O*-methylellagic acid (IC_50_ = 15.1 μM), and cereotagloperoxide (IC_50_ = 5.9 μM) (IC_50_ for BHA used as positive control was 44.2 μM) [44].

As shown in Table 6, similar to the DPPH test, the ABTS^•+^ assay revealed that the ethyl acetate fraction (CHs-4) of the leaves of *C. hsingshangensis* possessed the strongest ability to scavenge free radicals (IC_50_ = 0.37 ± 0.01 μg/mL), followed by the butanol fraction (CHs-3; IC_50_ = 0.68 ± 0.05 μg/mL) and the diethyl ether fraction (CHs-2; IC_50_ = 0.80 ± 0.02 μg/mL) of *C. hsingshangensis*. The ABTS^•+^ assay was also used to test different extracts of *C. nummularia*. The authors found that the greatest ABTS inhibition was caused by the methanol (IC_50_ = 0.020 mg/mL) and water extract (IC_50_ = 0.023 mg/mL) (IC_50_ for BHA used as a positive control was 0.015 mg/mL) [57]. Mahmutović-Dizdarević and co-authors [11] reported that IC_50_ values of methanolic extracts of leaves and barks of *C. horizontalis*, *C. integerrimus,* and *C. tomentosum* ranged from 0.12 (leaves of *C. tomentosum*) to 0.87 mg/mL (barks of *C. tomentosum*).

Phenolic compounds can reduce oxidative stress by several mechanisms that depend on their chemical structure. One of them is the chelation of metal ions, such as iron, which plays a key role in the production of damaging oxygen species [58]. 

The chelating ability was determined based on measurement of the percentage of inhibition of formation of the ferrozine-Fe^2+^ complex. As reported in Table 6, the extracts and fractions from leaves of both studied *Cotoneaster* species possessed the capacity to interfere with the formation of iron and ferrozine complexes, which suggests their high chelating capacity and ability to capture iron ions before ferrozine. The IC_50_ values of most of these extracts showed higher chelating activity than the positive control–Na_2_EDTA*2H_2_O (IC_50_ = 4.15 ± 0.10 µg/mL). Among the investigated extracts, the best activity was noticed in the ethyl acetate fraction of *C. hsingshangensis* (IC_50_ = 0.50 µg/mL). Zengin and co-authors [57] also evaluated the metal chelating capacity of different extracts of aerial parts of *C. nummularia*. They found that the highest chelating activity was expressed by the ethyl acetate extract (IC_50_ = 0.25 mg EDTA/g of extract). Among the ethanol extracts of 34 species belonging to Rosaceae, *Cotoneaster meyeri*, *C. morulus,* and *C. numullaria* caused moderate metal-chelating effects (5.91, 21.48, and 26.19 chelation%, respectively) [34]. 

#### 3.2.2. Enzyme Inhibitory Activity

Lipoxygenases (LOXs) are present in the human body and play a crucial role in the stimulation of inflammatory reactions. Exaggerated quantities of reactive oxygen species can induce inflammation that stimulates the release of cytokines and then the activation of lipoxygenases. They are connected with the spread of many diseases, and their inhibition is viewed as a relevant step in their prevention [59]. The lipoxygenase inhibitor screening assay kit is a popular method for lipoxygenase detection, which estimates the presence of hydroperoxides (4-hydroperoxy *cis-trans* 1,3-conjugated pentadienyl moiety within the unsaturated fatty acid) at various positions (5, 12-, 15-), which are produced in the lipoxygenation reaction using a purified lipoxygenase.

Sinha et al. [60] noted that perioxisome proliferator activated receptor (PRAR) ligands induce lipogenesis in cultured human sebocytes; hence, 5-lipoxygenase inhibitors demonstrate the ability to decline acne lesions, due to the fact that they possess the capability to reduce lipogenesis. The presence of PPARα receptors within sebocytes has been observed in peroxisomes, mitochondria, and microsomes.

The results of the inhibition of lipoxygenase are shown in Table 7. The diethyl ether fraction (CHs-2) and ethyl acetate fraction (CHs-4) of *C. hsingshangensis* showed a considerable ability to inhibit lipoxygenase activity (IC_50_ = 4.15, and 5.72 μg/mL, respectively), while the water fraction of *C. hissaricus* showed the lowest activity (IC_50_ = 129.46 μg/mL). Both CHs-2 and CHs-4 exhibited significantly higher inhibitory activity than that of nordihydroguaiaretic acid (NDGA) used as a positive standard (IC_50_ = 5.89 μg/mL). In a recent study of *Cotoneaster* species in regard to their capacity to inhibit lipoxygenase, it was found that different extracts of the leaves of *C. bullatus*, *C. integerrimus,* and *C. zabelii* have moderate activity with IC_50_ values in the range of 95.19 to 684.84 μg/mL. The best LOX inhibition was achieved by the butanol fraction of *C. bullatus* (IC_50_ = 95.19 μg/mL) [29]. The hydromethanolic extracts of fruits of nine *Cotoneaster* species cultivated in Poland showed the strongest inhibition of LOX, with IC_50_ values in the range of 62.54 (*C. zabelii*) to 165.76 (*C. nanshan*) μg/mL [28].

Skin is especially sensitive to reactive oxygen species because it is exposed to oxidative stress from both endogenous and exogenous sources. Although oxidative stress is a key factor in this process, hyaluronic acid also plays a notable role. Its integrity inside the dermal matrix is substantial for cell integrity and proliferation. Under oxidative stress, hyaluronidase, which is responsible for hyaluronic acid depolymerization, is overactivated and breaks down this anionic glycosaminoglycan, carrying on to the destruction of the proteoglycan system. This results in the deregulation of skin homeostasis and increases inflammatory and allergic conditions [61,62].

In our study, the ethyl acetate fraction of *C. hsingshangensis* exhibited the best hyaluronidase inhibition activity (IC_50_ = 1.89 μg/mL) compared to the other extracts (Table 7). Most importantly, such activity was even better than the positive standard—EGCG (IC_50_ = 6.25 μg/mL). In turn, the IC_50_ values for the crude extracts of *C. hissaricus* and *C. hsingshangensis* were 15.09 ± 0.61, and 6.82 ± 0.15 μg/mL, respectively. Previous studies have shown that methanol extracts of the leaves of *C. bullatus*, *C. integerrimus,* and *C. zabelii* reduced the activity of hyaluronidase in a concentration-dependent manner. The most active were the butanol fraction of *C. bullatus* and *C. zabelii* (IC_50_ = 2.81, and 6.08 μg/mL, respectively) and they had inhibition activity better than indomethacin used as a positive control (IC_50_ = 8.61 μg/mL) [29]. Moreover, methanol extracts of fruits of *C. bullatus*, *C. dielsianus*, *C. divaricatus*, *C. hjelmqvistii*, *C. horizontalis*, *C. lucidus*, *C. nanshan*, *C. splendens*, and *C. zabelii* showed moderate inhibition of LOX with IC_50_ values in the range of 25.65 (*C. lucidus*) to 45.64 μg/mL (*C. nanshan*) [28].

Cyclooxygenases catalyze two reactions, the first being a cyclooxygenase function consisting of the addition of molecular oxygen to arachidonic acid to form prostaglandin G_2_ (PGG_2_). The second is the conversion of PGG_2_ to PGH_2_ by a peroxidase function. Therefore, this enzyme performs the initial reaction in the arachidonic acid metabolic cascade, carrying on to the formation of pro-inflammatory, i.e., prostaglandins, which regulate smooth muscle contractility, platelet aggregation, and mediate pain. Cyclooxygenase constitutive (COX-1) is responsible for the maintenance of physiological prostanoid biosynthesis, and COX-2 (an inducible isoform) is connected to inflammatory cell types and tissues [63].

To determine the potential anti-inflammatory properties of *Cotoneaster* leaves, we also estimated the ability of the extracts and fractions to inhibit the conversion of arachidonic acid to PGH_2_ by ovine COX-1 and human recombinant COX-2 using a COX inhibitor screening assay kit (Cayman Chemical, MI, USA). As shown in Table 7, most of the studied extracts and fractions showed good activity against COX-1 and COX-2. The most active against COX-1 were CHs-2 (IC_50_ = 6.39 µg/mL) and CHs-4 (IC_50_ = 9.54 µg/mL), followed by CHi-2 (IC_50_ = 11.15 µg/mL) and CHs (IC_50_ = 13.02 µg/mL), while the weakest was fraction CHi-1 (IC_50_ = 62.54 µg/mL). Except for the water residual of *C. hissaricus*—CHi-1 (IC_50_ = 100.36 µg/mL) and ethyl acetate fraction of the *C. hissaricus*—CHi-4 (IC_50_ = 81.69 µg/mL), all extracts and fractions of both *Cotoneaster* species showed good activity against COX-2 (IC_50_ = 5.09–57.59 µg/mL).

#### 3.2.3. Antibacterial Activity

##### Diffusion Test in Solid Medium

The antibacterial activity of the *Cotoneaster* extracts and fractions was assessed by diffusion test in a solid medium and measuring the zones of bacterial growth inhibition. The larger the zone around the applied samples, the more active the extract/fraction is.

The data show (Figure 7, Appendix A) that the *Cotoneaster* fractions, not the starting crude extracts, possessed the strongest antimicrobial properties. The largest zones of growth inhibition of Gram-positive microaerobic bacteria were produced by CHs-2 fraction (23 mm–21 mm), then CHs-4 (19 mm–16 mm), CHs-1 (15 mm–14 mm), and CHs-3 fractions (14 mm–11 mm). These fractions against Gram-positive aerobic strains showed weaker activity (in the range 19 mm–6 mm) than against microaerobic bacteria. Crude *Cotoneaster* extracts CHi and CHs showed moderate activity against all tested Gram-positive strains (12 mm–6 mm). None of the samples were active against Gram-negative bacteria. Previous studies have reported the resistance of both Gram-positive and Gram-negative bacteria to the leaf and bark of *C. integerrimus*, *C. tomentosus*, and *C. horizontalis* methanolic extracts. Mahmutović-Dizdarević and co-authors [11] found that these extracts have significant antimicrobial activity against *Salmonella enterica*, *Pseudomonas aeruginosa*, *Escherichia coli*, Extended Spectrum Beta-Lactamase producing *E. coli* or ESBL *E. coli*, *Enterococcus faecalis*, *Staphylococcus aureus*, Methicillin-resistant *Staphylococcus aureus* or MRSA, and *Bacillus subtilis*. In a further study, the antibacterial activity of the ethanolic extract of roots of *C. acuminatus* was also investigated against *Bacillus pumilus*, *B. subtilis*, *E. coli*, *Microccocus glutamicus*, *S. aureus*, *Proteus vulgaris*, and *P. aeruginosa*. The greatest effect was found for 100 μg/mL extract, with growth inhibition zones 10–18 mm [64]. Using the broth microdilution method, the antibacterial activity of the water, methanol, and ethyl acetate extracts of *C. nummularia* was evaluated against *E. coli*, *Bacillus cereus*, *P. aeruginosa*, methicillin sensitive *S. aureus* (MSSA), *Klebsiella pneumoniae*, *Salmonella enteritidis*, *S. pneumoniae*, *Sarcina lutea*, *Enterococcus faecalis*, methicillin resistant *S. aureus* (MRSA), and strains of methicillin resistant *S. aureus* isolated from clinical samples. The authors found that *E. faecalis* was the most sensitive bacteria, and *B. cereus*, *K. pneumoniae*, and *S. enteritidis* were the most resistant bacteria against all extracts except for the ethyl acetate extract [57]. 

To the best of our knowledge, this study represents the first report of the antibacterial activity against *C. acnes*, *S. epidermidis*, and *C. granulosum* strains, which can be responsible for skin diseases of *Cotoneaster* species.

##### Results of MIC and MBC Determination

In order to determine the MIC values of the selected active fractions of crude *C. hsingshangensis* extract (CHi, CHs, CHs-1-CHs-4), an assay was performed using the double dilution method in a 96-well plate. After the plates were incubated under the appropriate conditions for the given strain, they were analyzed in an automatic plate reader (at 600 nm) against the control. The concentration at which no bacterial growth was observed was taken as the MIC. Next, the MBC value was determined by dispensing 10 µL of the solutions taken from the wells of 96-well plates in which no bacterial growth was observed.

The data contained in Table 8 confirm the results obtained in the earlier diffusion assay; namely, the tested fractions are the most active against the acne strains of *Cutibacterium* spp.

The most favorable, i.e., the lowest MIC values against these bacteria, were obtained by fractions CHs-2 (MIC 31.25–125 µg/mL), CHs-4 (125–250 µg/mL), CHs-1, and CHs-3 (both 500–1000 µg/mL).

Next, the MBC—as the lowest concentration of an antibacterial sample required to kill the bacterium—can be assayed based on an MIC test by subculturing samples on agar plates. Thus, the clear medium from MIC plates was spread on fresh agar to check its sterility. The concentration of the sample that did not produce colonies was considered the MBC power. According to the ratio MBC/MIC, we visualized and appreciated antibacterial activity. If the ratio MBC/MIC ≤ 4, the effect was considered bactericidal, but if the ratio MBC/MIC > 4, the effect was defined as bacteriostatic. Data of MBC/MIC ratio displayed in Table 8 show that none of *C. hsingshangensis* fractions showed any bactericidal activity in relation to the acne bacteria. The most promising power against microaerobic Gram-positive strains was displayed for CHs-2 (diethyl ether fraction of *C. hsingshangensis*). The remaining extract power (CHs-1, CHs-3, CHs-4) was not measurable.

#### 3.2.4. Cytotoxic Activity

The obtained data revealed that all *Cotoneaster* extracts possessed low cytotoxicity toward normal human fibroblasts (Figure 8), as it was not possible to determine their CC_50_ values at tested concentrations (1.95–1000 μg/mL). The CC_50_ value denotes the extract concentration that inhibits BJ viability to 50%. Among the tested *Cotoneaster* extracts, CHs-3 exhibited the lowest ability to inhibit the viability of BJ cells because at the highest tested concentration (1000 μg/mL), the cell viability was 77.06 ± 1.57%. For comparison, fibroblast viability treated with CHi, CHs, CHs-1, and CHs-4 at the same concentration was 60.60 ± 1.47%, 53.18 ± 1.84%, 68.28 ± 5.55%, and 63.80 ± 4.39%, respectively. In the case of CHs-2, cell viability after incubation at concentrations of 250 μg/mL, 500 μg/mL, and 1000 μg/mL was 54.40 ± 3.56%, 43.35 ± 2.02%, and 45.89 ± 4.57%, respectively. Thus, these results may suggest that the CC_50_ value for this substance should be detected in the range 250–500 μg/mL. As a consequence, it was demonstrated that CHs-2 possessed the highest ability to inhibit BJ cell viability compared to other substances.

### 3.3. Multivariate Analysis of the Results

To perform a holistic view of the results, we used a chemometric multivariate approach. It allows us to examine some independent trends in changes among the investigated properties. They were investigated using hierarchical cluster analysis (HCA, Figure 9) and principal component analysis (PCA, Figure 10).

The analysis was done on a column-wise scaled matrix, as each parameter is ex-pressed with different units. Additionally, two of the parameters, DPPH and ABTS, were expressed as negative values in this matrix. This was done to convert the strong correlation between them and TPC, TPAC, and TFC from negative to positive. This places the loading vectors in the same direction and allows us to see everything visually in more detail.

It can be clearly seen that the parameters form two distinct groups, expressed as two separate groups of loading arrows and two main branches of the dendrogram. The first group contains antioxidant parameters TPC, TPAC, and TFC, together with DPPH and ABTS. The other parameters form the second group. 

The increase or decrease of all investigated parameters in opposite directions is responsible for 75% of total variance and is modeled as PC1. PC2 represents an intercorrelated increase or decrease of all investigated parameters and is responsible for 14% of the variability. Further PCs do not contain any interpretable information (results not shown).

## 4. Conclusions

The management of acne vulgaris should be “multidirectional”. Therapeutics should reduce sebum production, as well as follicular hyperkeratinization of the epidermal cells. For this reason, they should primarily exhibit antioxidant, anti-inflammatory, and antimicrobial activities, without cytotoxic effects.

Our analysis has been designed due to missing data in the available literature, namely considerable gaps in the knowledge considering *C. hissaricus* and *C. hsingshangensis* are observed. The unique chemical composition of the representatives of the Rosaceae family makes it possible to obtain a wide range of pharmaceutical, medicinal, and cosmetic products. Thus, *Cotoneaster* species seem to be promising candidates for further research. Chemical drugs have limitations because of their high toxic activity, as well as their ability to induce adverse side effects. Thus, a growing interest in the use of herbal medicines for the management of acne and other diseases is increasingly recognized.

Among approximately 2500 species belonging to the family Rosaceae, in vitro anti-acne research has been carried out only with a small number of them, namely *Rosa damascene* [65], *Rosa multiflora* [66], *Prunus jamasakura* [67], and apple polyphenols [54]. Interestingly, there is a lack of research on other plants from the *Rosaceae* family for anti-acne activity and the effect of phytochemicals on the skin. Simultaneously, indisputable evidence that phenolic compounds exhibit the desired effects and contribute to alleviate skin disorders can be found in the available literature [51,68]. Furthermore, clindamycin and kaempferol in combination with quercetin possess more beneficial effects than other formulations [69].

In our in vitro research, we characterized composition and evaluated the biological properties of extracts from *C. hissaricus* and *C. hsingshangensis*. Thus, we identified the main compounds present in extracts, as well as determined the antioxidant, anti-inflammatory, antimicrobial, and cytotoxic properties of such extracts.

To sum up our results, it seems to be clear that comprehensive and well-designed future research on phenolic compounds from *Cotoneaster* in greater detail will constitute significant importance in pharmacy and medicine. Among the studied extracts, the diethyl ether fraction of *C. hsingshangensis* (CHs-2) exhibited great ability to scavenge free radicals and good capacity to inhibit cyclooxygenase-1, cyclooxygenase-2, lipoxygenase, and hyaluronidase. Moreover, it had the most promising power against microaerobic Gram-positive strains, and importantly, it was non-toxic toward normal skin fibroblasts. Taking into account the value of the calculated therapeutic index (>10), it is worth noting that CHs-2 can be subjected to in vivo study and constitutes a promising anti-acne agent.

To the best of our knowledge, there is no reported study on the chemical composition and skin-related properties of the examined species.

## Figures and Tables

**Figure 1 cells-11-00367-f001:**
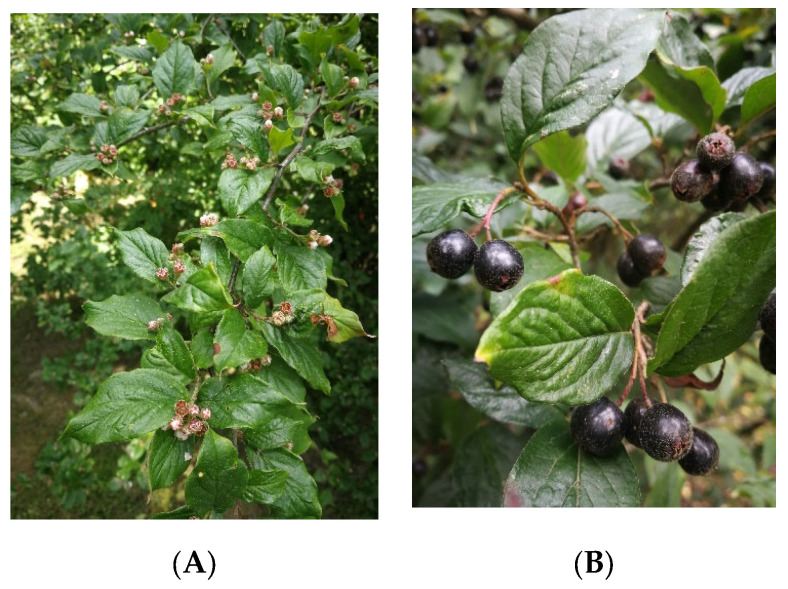
Photos of *C. hsingshangensis* under study collected in: (**A**) May, (**B**,**C**) September, and (**D**) October.

**Figure 2 cells-11-00367-f002:**
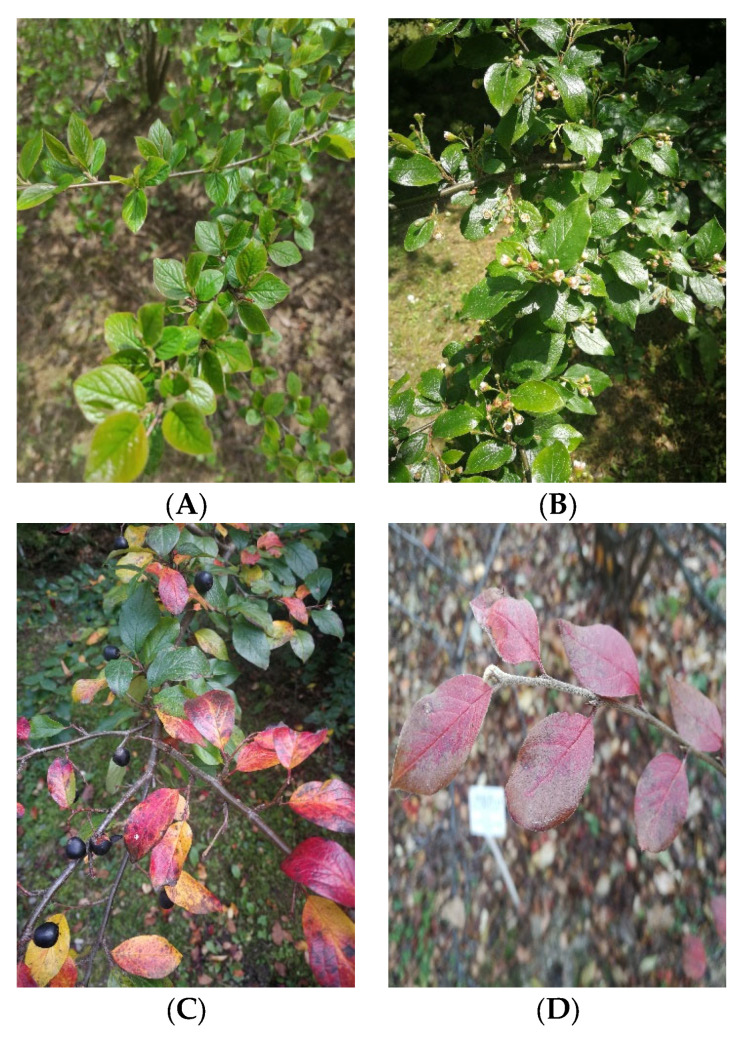
Photos of *C. hissaricus* under study were collected in (**A**) April, (**B**) May, (**C**) September, and (**D**) October.

**Figure 3 cells-11-00367-f003:**
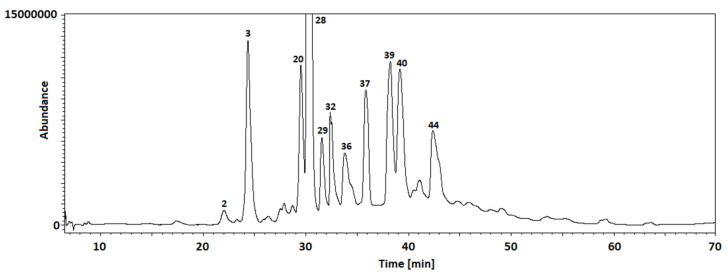
TIC chromatogram of *C. hissaricus* extract.

**Figure 4 cells-11-00367-f004:**
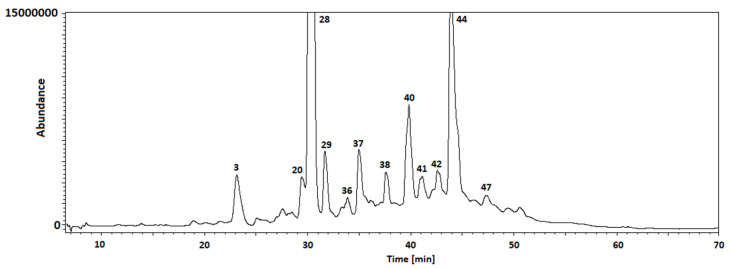
TIC chromatogram of *C. hsingshangensis* extract.

**Figure 5 cells-11-00367-f005:**
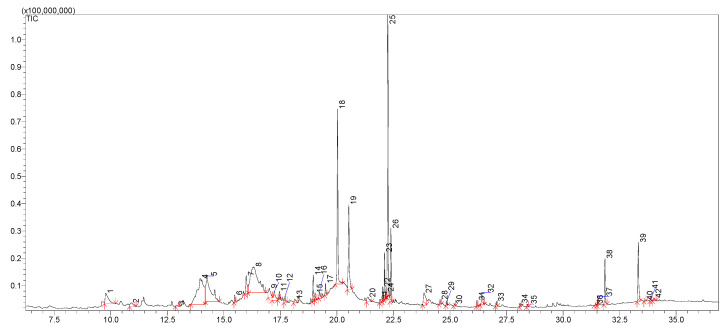
GC/MS chromatogram (in TIC mode) of *C. hissaricus* extract.

**Figure 6 cells-11-00367-f006:**
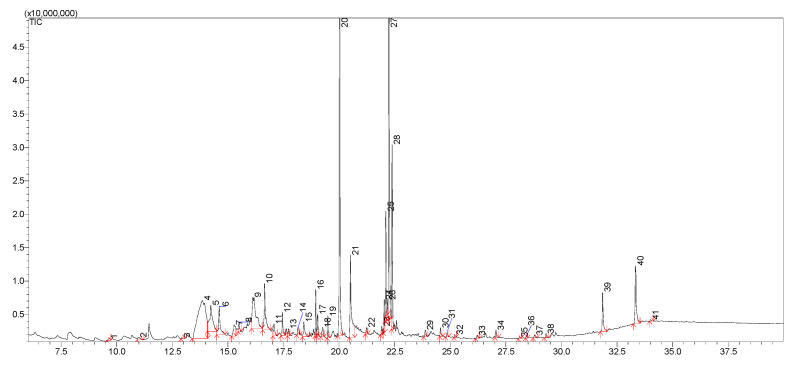
GC/MS chromatogram (in TIC mode) of *C. hsingshangensis* extract.

**Figure 7 cells-11-00367-f007:**
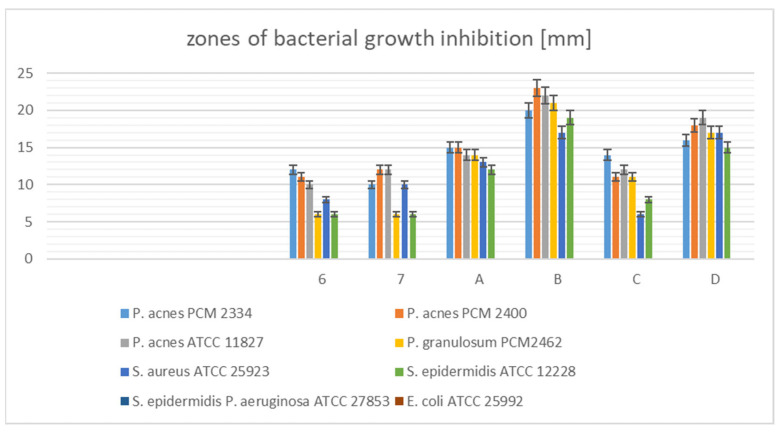
Zones of bacterial growth inhibition of the *Cotoneaster* extracts.

**Figure 8 cells-11-00367-f008:**
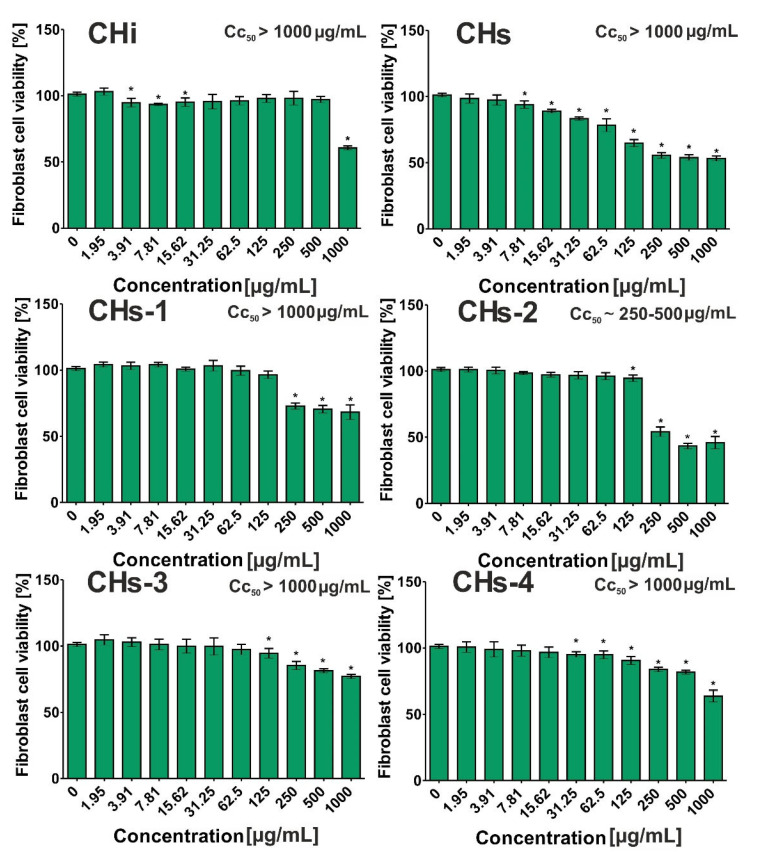
Cytotoxic effect of *Cotoneaster* extracts on human normal fibroblasts. (BJ cell line, ATCC CRL-2522TM). The cell viability was assessed after 24-h incubation using the MTT assay. Asterisk (*) denotes significantly different data (*p* < 0.05, unpaired *t*-test) compared to the control, namely culture medium without substances—0 μg/mL.

**Figure 9 cells-11-00367-f009:**
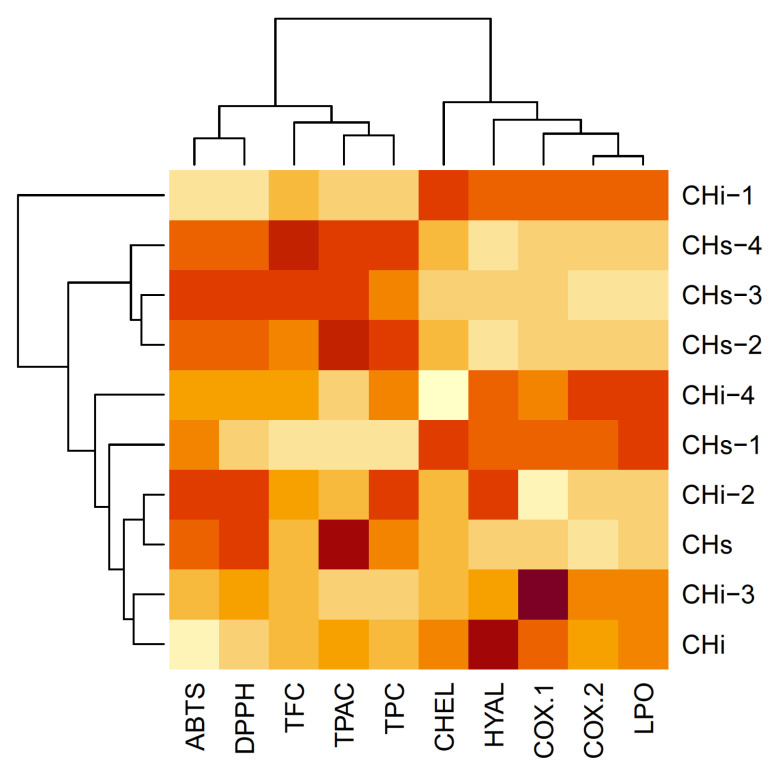
The heatmap analysis of the scaled matrix, with dendrograms based on Euclidean distance. TPC—Total phenolic content; TPCA—Total phenolic acids content; TFC—Total flavonoid content; CHEL—Metal chelating activity; LPO—Lipoxygenase inhibition; HYAL—Hyaluronidase inhibition.

**Figure 10 cells-11-00367-f010:**
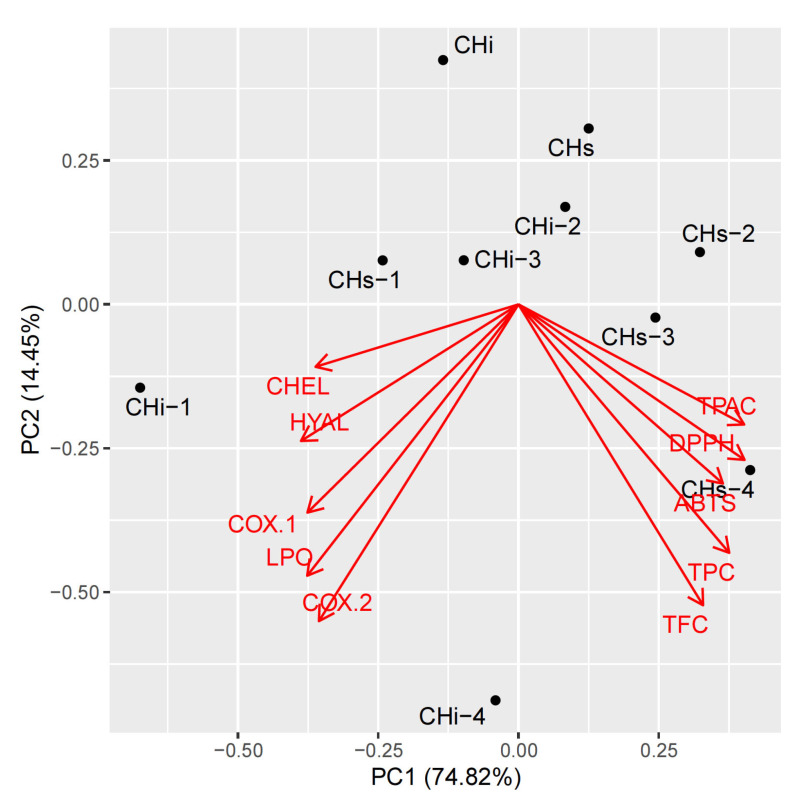
Principal component analysis loadings of the investigated dataset: PC1 vs. PC2 plot. TPC—Total phenolic content; TPCA—Total phenolic acids content; TFC—Total flavonoid content; CHEL—Metal chelating activity; LPO—Lipoxygenase inhibition; HYAL—Hyaluronidase inhibition.

**Table 1 cells-11-00367-t001:** The total content of phenolic (TPC), flavonoid (TFC), and phenolic acids (TPAC) in the *C. hissaricus* and *C. hsingshangensis* leaf extracts.

Sample	Extraction Yield (% DE)	Total Phenolic Content [mg GAE/g DE]	Total Phenolic Acids [mg CAE/g DE]	Total Flavonoid Content [mg QE/g DE]
CHi	15.30	193.84 ± 1.14 ^e,f,g,h,i^	33.80 ± 1.03 ^e,f,g,h,i^	25.38 ± 0.35 ^e,f,g,h,i^
CHi-1	3.00	83.87 ± 0.23 ^a,e,f,g,h,i^	10.31 ± 0.11 ^a,e,f,g,h,i^	0.12 ± 0.05 ^a,e,g,h,i^
CHi-2	0.89	348.05 ± 2.81 ^a,b,e,f,g,h,i^	40.57 ± 0.19 ^a,b,e,f,g,h,i^	72.12 ± 0.35 ^a,b,e,f,g,h,i^
CHi-3	17.29	219.00 ± 0.49 ^a,b,c,e,f,g,h,i^	37.83 ± 0.15 ^a,b,c,e,f,g,h,i^	58.71 ± 0.49 ^a,b,c,e,f,g,h,i^
CHi-4	6.86	463.16 ± 3.94 ^a,b,c,d,e,f,g,h,i^	56.12 ± 0.27 ^a,b,c,d,e,f,g,h^	134.89 ± 0.51 ^a,b,c,d,i^
CHs	27.22	296.13 ± 1.52	61.27 ± 0.93	47.72 ± 0.37
CHs-1	6.73	143.30 ± 1.09 ^e^	19.65 ± 0.30 ^e^	0.23 ± 0.02 ^e^
CHs-2	2.07	486.04 ± 3.17 ^e,f^	83.94 ± 0.25 ^e,f^	97.66 ± 0.18 ^e,f^
CHs-3	14.04	381.83 ± 2.53 ^e,f,g^	70.17 ± 0.13 ^e,f,g^	146.58 ± 1.10 ^e,f,g^
CHs-4	5.76	559.77 ± 3.76 ^e,f,g,h^	91.95 ± 0.48 ^e,f,g,h^	252.27 ± 0.24 ^e,f,g,h^

DE—dry extract; GAE—Gallic Acid Equivalent; CAE—Caffeic Acid Equivalent; QE—Quercetin Equivalent; CHi—methanol–acetone–water (3:1:1, *v/v*) extract of *C. hissaricus*; CHs—methanol–acetone–water (3:1:1, *v/v*) extract of *C. hsingshangensis*; CHi-1—water fraction of *C. hissaricus*, CHi-2—diethyl ether fraction of *C. hissaricus*, CHi-3—butanol fraction of *C. hissaricus*, CHi-4—ethyl acetate fraction of *C. hissaricus;* CHs-1—water fraction of *C. hsingshangensis*, CHs-2—diethyl ether fraction of *C. hsingshangensis*, CHs-3—butanol fraction of *C. hsingshangensis*, CHs-4—ethyl acetate fraction of *C. hsingshangensis*. Values were presented as mean ± standard deviation (*n* = 9). Statistical analysis: a—significantly different results compared to CHi; b—significantly different results compared to CHi-1; c—significantly different results compared to CHi-2; d—significantly different results compared to CHi-3; e—significantly different results compared to CHs; f—significantly different results compared to CHs-1; g—significantly different results compared to CHs-2; h—significantly different results compared to CHs-3; i—significantly different results compared to CHs-4; One-Way ANOVA test, followed by a Tukey’s multiple comparison test, *p* < 0.05.

**Table 2 cells-11-00367-t002:** High-resolution mass spectrometry (HR-MS) of [M − H]^−^ ion and MS2 data.

Peak No.	Name of Compound	[M − H]^−^	MS2	Theoretical Mass [M − H]^−^ (Da)	Experimental Mass [M − H]^−^ (Da)	Δ ppm	Δ mDa	Elemental Composition
1	Mannitol	181	59,71,73,85,89,101,113,119,163	181.07122	181.07118	0.22	−0.04	C_6_H_13_O_6_
4	Quercetin-3-*O*-(2″-*O*-xylosyl)galactoside	595	271,301b,435	595.12992	595.12999	0.12	0.07	C_26_H_27_O_16_
5	Quercetin-3-*O*-gentiobioside	625	271,301b,463	625.14048	625.14038	0.16	−0.10	C_27_H_29_O_17_
6	Vitexin-2″-*O*-arabinoside	563	283,432b	563.14009	563.14001	0.14	−0.08	C_26_H_27_O_14_
7	Apigenin-6,8-*C*-dicelobioside	593	325,386b,387	593.15065	593.15077	0.20	0.12	C_27_H_29_O_15_
8	Vitexin-2″-*O*-rhamnoside	577	283,432b	577.15574	577.15571	0.05	−0.03	C_27_H_29_O_14_
9	Quercetin-3-*O*-glucoside (Isoquercitrin)	463	271,300b,301	463.08766	463.08775	0.19	0.09	C_21_H_19_O_12_
10	Quercetin-3-*O*-galactoside (Hyperoside)	463	271,300b,301	463.08766	463.08763	0.06	−0.03	C_21_H_19_O_12_
11	Kaempferol-3-*O*-glucoside (Astragalin)	447	284b,300	447.09274	447.09289	0.34	0.15	C_21_H_19_O_11_
12	Quercetin-3-*O*-rhamnoside (Quercitrin)	447	284b,300	447.09274	447.09271	0.07	−0.03	C_21_H_19_O_11_
13	7-Methylkaempferol-4′-*O*-glucoside	461	298b,315	461.10839	461.10849	0.22	0.10	C_22_H_21_O_11_
14	3′,4′-dihydroxy-6-methoxyflavone-7-*O*-rhamnoside	429	255,283b,400,401	429.11856	429.11853	0.07	−0.03	C_22_H_21_O_9_
15	Apigenin-8-*C*-glucoside (Vitexin)	431	283,311b,312,341	429.11856	429.11842	0.33	−0.14	C_21_H_19_O_10_
16	Apigenin-7-*O*-glucoside	447	269b,270,431	431.09783	431.09797	0.32	0.14	C_21_H_19_O_10_
17	Biochanin A-7-*O*-glucoside (Sissotrin)	445	132,211,223,224,239b,240,267	445.11348	445.11355	0.16	0.07	C_22_H_21_O_10_
18	5,7,2′,5′-tetrahydroxyflavanone-7-*O*-glucoside	449	286b,302	449.10839	449.10825	0.31	−0.14	C_21_H_21_O_11_
19	5-Methylgenistein-4′-*O*-glucoside	445	255,283b,417,430	445.11348	445.11339	0.20	−0.09	C_22_H_21_O_10_
20	Gentisic acid 2-*O*-glucoside (Orbicularin)	315	153b,271	315.07161	315.07149	0.38	−0.12	C_13_H_15_O_9_
27	Caffeoylmalic acid	295	133,135,179b	295.0454	295.04533	0.24	−0.07	C_13_H_11_O_8_
28	Prunasin	294	161b	294.09777	294.09771	0.20	−0.06	C_14_H_16_NO_6_
29	Amygdalin	456	89,119,143,179,221,263,323b	456.15059	456.15047	0.26	−0.12	C_20_H_26_NO_11_
33	7,8-dimethoxy-6-hydroxycoumarin	221	191,206b,207	221.045	221.04506	0.27	0.06	C_11_H_9_O_5_
35	Cotonoate A	291	121,165,247b	291.15964	291.15975	0.38	0.11	C_17_H_23_O_4_
36	Horizontoate A	263	148,164,219b	263.12834	263.12852	0.68	0.18	C_15_H_19_O_4_
37	3,3′,4′-tri-*O*-methylellagic acid	343	255,284,299b,329	343.0454	343.04554	0.41	0.14	C_17_H_11_O_8_
40	Scopoletin	191	104,105,120,148b,176	191.03444	191.03439	0.26	−0.05	C_10_H_7_O_4_
41	Arbutin	271	71,101,108b,109,113,161	271.08178	271.08175	0.11	−0.03	C_12_H_15_O_7_
42	5-Methylgenistein	283	239b,255,269	283.06065	283.06053	0.42	−0.12	C_16_H_11_O_5_
44	Eriodictyol	287	151,162b	287.05557	287.05551	0.21	−0.06	C_15_H_11_O_6_
45	5,7,2′,5′-tetrahydroxyflavanone	287	151,162b,259	287.05557	287.05564	0.24	0.07	C_15_H_11_O_6_
46	Naringenin	271	119,151b,187	271.06065	271.06061	0.15	−0.04	C_15_H_11_O_5_
47	Horizontoate C	482	210,272b,288	482.42093	482.42097	0.08	0.04	C_29_H_56_NO_4_

b—base peak.

**Table 3 cells-11-00367-t003:** Content of active compounds in the leaves of *C. hissaricus* (CHi) and *C. hsingshangensis* (CHs).

No	Compound	Calibration Standard	Amounts [μg/g DE]	
			CHi	CHs
1	mannitol	glucose	6834 ± 249	3104 ± 123 *
2	ascorbic acid	ascorbic acid	298 ± 10	1726 ± 66 *
3	quercetin 3-*O*-rutinoside (rutin)	rutin	18,028 ± 650	3823 ± 131*
4	quercetin 3-*O*-(2″-*O*-xylosyl)galactoside	rutin	7318 ± 289	2667 ± 99 *
5	quercetin 3-*O*-gentiobioside	rutin	2759 ± 109	2310 ± 87
6	vitexin 2″-*O*-arabinoside	rutin	5625 ± 233	3249 ± 121 *
7	apigenin 6,8-*C*-dicelobioside	rutin	5926 ± 225	1273 ± 50 *
8	vitexin 2″-*O*-rhamnoside	rutin	3923 ± 154	1268 ± 49 *
9	quercetin 3-*O*-glucoside (isoquercitrin)	rutin	10,079 ± 353	8926 ± 327 *
10	quercetin 3-*O*-galactoside (hyperoside)	rutin	9119 ± 331	6184 ± 237 *
11	kaempferol 3-*O*-glucoside (astragalin)	rutin	2480 ± 92	4430 ± 168 *
12	quercetin 3-*O*-rhamnoside (quercitrin)	rutin	2158 ± 80	6726 ± 251 *
13	7-methylkaempferol 4′-*O*-glucoside	rutin	3939 ± 137.9	2079 ± 79 *
14	3′,4′-dihydroxy-6-methoxyflavone 7-*O*-rhamnoside	rutin	1538 ± 50	443 ± 16 *
15	apigenin 8-*C*-glucoside (vitexin)	rutin	2454 ± 94	1930 ± 77 *
16	apigenin 7-*O*-glucoside	rutin	1486 ± 57	3734 ± 149 *
17	biochanin A 7-*O*-glucoside (sissotrin)	rutin	724 ± 26	709 ± 23
18	5,7,2′,5′-tetrahydroxyflavanone 7-*O*-glucoside	rutin	8067 ± 290	2469 ± 93 *
19	5-methylgenistein 4′-*O*-glucoside	rutin	1205 ± 43	943 ± 34 *
20	orbicularin	quercetin	3838 ± 154	1818 ± 79 *
21	*p*-hydroxybenzoic acid	*p*-hydroxybenzoic acid	897 ± 30	823 ± 27 *
22	benzoic acid	benzoic acid	339 ± 12	441 ± 16
23	gentisic acid	gentisic acid	258 ± 10	<LOQ
24	protocatechuic acid	protocatechuic acid	509 ± 20	69 ± 3 *
25	syringic acid	syringic acid	753 ± 31	1130 ± 39 *
26	vanillic acid	vanillic acid	<LOQ	<LOQ
27	caffeoylmalic acid	caffeic acid	2828 ± 109	1459 ± 51 *
28	chlorogenic acid	chlorogenic acid	37,932 ± 1330	60,043 ± 2231 *
29	prunasin	glucose	3360 ± 127	1226 ± 61
30	*p*-coumaric acid	*p*-coumaric acid	329 ± 13	539 ± 19 *
31	amygdalin	glucose	1803 ± 63	555 ± 19 *
32	caffeic acid	caffeic acid	6118 ± 223	2220 ± 81 *
33	cinnamic acid	cinnamic acid	924 ± 31	2429 ± 89 *
34	ferulic acid	ferulic acid	644 ± 24	1567 ± 65 *
35	salicylic acid	salicylic acid	657 ± 27	823 ± 30 *
36	7,8-dimethoxy-6-hydroxycoumarin	umbelliferone	1209 ± 43	<LOQ
37	cotonoate A	benzoic acid	1564 ± 55	<LOQ
38	horizontoate A	benzoic acid	<LOQ	<LOQ
39	3,3′,4′-tri-*O*-methylellagic acid	quercetin	2053 ± 81	1169 ± 45 *
40	scopoletin	umbelliferone	12,219 ± 440	10,481 ± 371 *
41	arbutin	glucose	1126 ± 41	1084 ± 46 *
42	5-methylgenistein	quercetin	1109 ± 39	1423 ± 59 *
43	quercetin	quercetin	229 ± 7	115 ± 4 *
44	horizontoate C	oleic acid	5516 ± 219	2939 ± 119 *
45	eriodictyol	quercetin	59 ± 3	<LOQ
46	5,7,2′,5′-tetrahydroxyflavanone	quercetin	89 ± 5	<LOQ
47	naringenin	quercetin	2627 ± 97	2121 ± 88 *

LOQ—limit of quantification; DE—dry extract. Asterisk (*) denotes significantly different data (*p* < 0.05, unpaired *t*-test) compared to CHi.

**Table 4 cells-11-00367-t004:** Composition of the extracts of leaves of *C. hissaricus* (% of total fraction; mass%, GC).

No.	tr	Area	%Area	Compound
1	9.776	39557931	2.52	Benzofuran, 2,3-dihydro-
2	10.908	12341078	0.79	Isosorbide
3	12.913	3452024	0.22	Furan, 3-(4-methyl-5-*trans*-phenyl-1,3-oxazolidin-2-yl)-
4	13.952	192528086	12.29	1,3:2,5-Dimethylene-L-rhamnitol
5	14.242	135536063	8.65	d-Glycero-d-galacto-heptose
6	15.48	7430877	0.47	Megastigmatrienone
7	15.988	32418863	2.07	3-Hydroxy-.beta.-damascone
8	16.329	229108049	14.62	2-Deoxy-d-galactose
9	17.009	21401978	1.37	2-Hydroxyhexadecyl butanoate
10	17.229	21671810	1.38	(E)-2,6-Dimethoxy-4-(prop-1-en-1-yl)phenol
11	17.448	5862283	0.37	Myristic acid methyl ester
12	17.709	6588889	0.42	9-(3,3-Dimethyloxiran-2-yl)-2,7-dimethylnona-2,6-dien-1-ol
13	18.137	7027919	0.45	6-Hydroxy-4,4,7a-trimethyl-5,6,7,7a-tetrahydrobenzofuran-2(4H)-one
14	18.951	19840196	1.27	Neophytadiene
15	19.034	6060805	0.39	2-Pentadecanone, 6,10,14-trimethyl-
16	19.222	10207664	0.65	[1,1′-Bicyclopropyl]-2-octanoic acid, 2′-hexyl-, methyl ester
17	19.496	10733207	0.69	3,7,11,15-Tetramethyl-2-hexadecen-1-ol
18	20.028	157760294	10.07	Hexadecanoic acid, methyl ester
19	20.525	95409334	6.09	Palmitic acid
20	21.372	11475712	0.73	Cyclohexanebutanoic acid
21	21.913	3242083	0.21	Methyl *trans*-4-(2-nonylcyclopentyl)butanoate
22	22.029	10523560	0.67	*cis*-11,14-Eicosadienoic acid, methyl ester
23	22.104	33503994	2.14	Linolenic acid methyl ester
24	22.17	5141879	0.33	9-Octadecenoic acid (Z)-, methyl ester
25	22.251	247573718	15.79	Phytol, acetate
26	22.387	53091621	3.39	Methyl stearate
27	23.854	32049624	2.05	Benzyl.beta.-d-glucoside
28	24.561	3681746	0.24	Methyl 18-methylnonadecanoate
29	24.852	4060228	0.26	4,8,12,16-Tetramethylheptadecan-4-olide
30	25.205	2015357	0.13	3,7,11,15-Tetramethylhexadec-2-en-1-yl acetate
31	26.197	5666720	0.36	1-Heptacosanol
32	26.403	17982618	1.15	1-Docosanol, methyl ether
33	27.053	3938497	0.25	Phytyl palmitate
34	28.102	4304975	0.27	9-Hexacosene
35	28.421	2176024	0.14	Methyl 18-methylnonadecanoate
36	31.436	1813814	0.12	Cholesta-4,6-dien-3-ol, (3.beta.)-
37	31.551	8162207	0.52	Octatriacontyl trifluoroacetate
38	31.851	40546560	2.59	dl-.alpha.-Tocopherol
39	33.331	54433136	3.47	beta.-Sitosterol
40	33.63	1502659	0.10	Acetyl betulinaldehyde
41	33.87	1185800	0.08	Urs-12-en-28-al
42	34.019	3680922	0.23	Acetyl betulinaldehyde

**Table 5 cells-11-00367-t005:** Composition of the extracts of leaves of *C. hsingshangensis* (% of total fraction; mass%, GC).

No.	tr	Area	%Area	Compound
1	9.655	1862168	0.19	Benzofuran, 2,3-dihydro-
2	10.990	2840045	0.28	Isosorbide
3	12.933	560231	0.06	Furan, 3-(4-methyl-5-*trans*-phenyl-1,3-oxazolidin-2-yl)-
4	13.854	146558815	14.62	1,3:2,5-Dimethylene-L-rhamnitol
5	14.235	38707499	3.86	d-Glycero-d-galacto-heptose
6	14.608	21753970	2.17	Lauric acid methyl ester
7	15.280	11348450	1.13	Tridecanoic acid
8	15.501	4033264	0.40	Megastigmatrienone
9	16.125	55318375	5.52	2-Deoxy-d-galactose
10	16.648	30284734	3.02	d-Glucitol, 2,5-anhydro-1-*O*-octyl-
11	17.070	6971059	0.70	2-Hydroxyhexadecyl butanoate
12	17.456	8567378	0.85	Myristic acid methyl ester
13	17.725	2962331	0.30	9-(3,3-Dimethyloxiran-2-yl)-2,7-dimethylnona-2,6-dien-1-ol
14	18.153	4592151	0.46	6-Hydroxy-4,4,7a-trimethyl-5,6,7,7a-tetrahydrobenzofuran-2(4H)-one
15	18.416	10513893	1.05	Benzenesulfonamide, N-butyl-
16	18.952	13881898	1.38	Neophytadiene
17	19.037	7043508	0.70	2-Pentadecanone, 6,10,14-trimethyl-
18	19.269	4920685	0.49	[1,1′-Bicyclopropyl]-2-octanoic acid, 2′-hexyl-, methyl ester
19	19.500	7077957	0.71	3,7,11,15-Tetramethyl-2-hexadecen-1-ol
20	20.027	170606966	17.02	Hexadecanoic acid, methyl ester
21	20.515	51495640	5.14	Palmitic acid
22	21.239	2788195	0.28	Cyclohexanebutanoic acid
23	21.920	3248248	0.32	Methyl *trans*-4-(2-nonylcyclopentyl)butanoate
24	22.033	9779382	0.98	*cis*-11,14-Eicosadienoic acid, methyl ester
25	22.105	34030060	3.39	Linolenic acid methyl ester
26	22.172	3962623	0.40	9-Octadecenoic acid (Z)-, methyl ester
27	22.247	216221222	21.55	Phytol, acetate
28	22.386	56126780	5.60	Methyl stearate
29	23.876	3905372	0.39	Benzyl.beta.-d-glucoside
30	24.565	3540084	0.35	Methyl 18-methylnonadecanoate
31	24.857	1679444	0.17	4,8,12,16-Tetramethylheptadecan-4-olide
32	25.213	1068195	0.11	3,7,11,15-Tetramethylhexadec-2-en-1-yl acetate
33	26.211	1769239	0.18	1-Heptacosanol
34	27.057	2638778	0.26	Phytyl palmitate
35	28.117	1110457	0.11	9-Hexacosene
36	28.430	1074607	0.11	Methyl 18-methylnonadecanoate
37	28.792	1507426	0.15	Neophytadiene
38	29.305	1886233	0.19	2,6,10,14,18-Pentamethyl-2,6,10,14,18-eicosapentaene
39	31.856	19787504	1.97	dl-α-Tocopherol
40	33.332	33729826	3.36	β-Sitosterol
41	34.009	802361	0.08	Acetyl betulinaldehyde

**Table 6 cells-11-00367-t006:** The IC_50_ values determined in antioxidant tests.

Sample	IC_50_		
	DPPH [μg/mL]	ABTS [μg/mL]	CHEL [μg/mL]
CHi	21.73 ± 0.13 ^e,f,g,h,i,k,l^	4.14 ± 0.10 ^e,f,g,h,i,j,k,l^	3.54 ± 0.15 ^f^
CHi-1	32.37 ± 0.19 ^a,e,f,g,h,i,k,l^	5.48 ± 0.12 ^a,e,f,g,h,i,j,k,l^	182.67 ± 4.19 ^a,e,f,g,h,i,m^
CHi-2	8.83 ± 0.10 ^a,b,e,f,g,h,i,j,k,l^	1.68 ± 0.03 ^a,b,e,f,g,h,i,k,l^	2.94 ± 0.03 ^b,f^
CHi-3	13.69 ± 0.14 ^a,b,c,e,f,g,h,i,j,k,l^	2.74 ± 0.11 ^a,b,c,e,f,g,h,i,j,k,l^	4.04 ± 0.16 ^b,f^
CHi-4	5.40 ± 0.10 ^a,b,c,d,e,f,g,h,i,j,k,l^	0.90 ± 0.02 ^a,b,c,d,e,f,i,j,k,l^	1.19 ± 0.05 ^b,f,i,m^
CHs	10.16 ± 0.02 ^j,k,l^	1.92 ± 0.13 ^j,l^	1.73 ± 0.10 ^j,m^
CHs-1	18.15 ± 0.13 ^e,j,k,l^	1.53 ± 0.09 ^e,j^	76.50 ± 1.45 ^e,j,m^
CHs-2	4.15 ± 0.05 ^e,f,j,k,l^	0.80 ± 0.02 ^e,f,j,k,l^	1.10 ± 0.03 ^f,j,m^
CHs-3	3.43 ± 0.02 ^e,f,g,j,k^	0.68 ± 0.05 ^e,f,j,k,l^	1.01 ± 0.01 ^f,j,m^
CHs-4	2.08 ± 0.03 ^e,f,g,h,k,l^	0.37 ± 0.01 ^e,f,g,h,j,k,l^	0.50 ± 0.01 ^f,j,m^
quercetin	2.05 ± 0.10	3.27 ± 0.23	6.24 ± 0.19
AA	4.75 ± 0.16	1.73 ± 0.09	nt
Trolox	3.68 ± 0.09	1.31 ± 0.05	nt
Na_2_EDTA*2H_2_O	nt	nt	4.15 ± 0.10

Data were expressed as mean ± SD, *n* = 3. AA—ascorbic acid; Na_2_EDTA*2H_2_O—ethylenediaminetetraacetic acid, disodium dihydrate; nt—not tested; CHi—crude extract of *C. hissaricus*, CHs—crude extract of *C. hsingshangensis*, CHi-1/CHs-1—water fraction, CHi-2/CHs-2—diethyl ether fraction, CHi-3/CHs-3—butanol fraction, CHi-4/CHs-4—ethyl acetate fraction; ABTS—2,2′-azino-bis-(3-ethyl-benzothiazole-6-sulfonic acid); CHEL—metal chelating activity. Statistical analysis: a—significantly different results compared to CHi; b—significantly different results compared to CHi-1; c—significantly different results compared to CHi-2; d—significantly different results compared to CHi-3; e—significantly different results compared to CHs; f—significantly different results compared to CHs-1; g—significantly different results compared to CHs-2; h—significantly different results compared to CHs-3; i—significantly different results compared to CHs-4; j—significantly different results compared to quercetin; k—significantly different results compared to AA; l—significantly different results compared to Trolox; m—significantly different results compared to Na_2_EDTA*2H_2_O; One-Way ANOVA test, followed by a Tukey’s multiple comparison test, *p* < 0.05.

**Table 7 cells-11-00367-t007:** Anti-lipoxygenase, anti-hyaluronidase, and anti-cyclooxygenase activities of the leaves of *C. hissaricus* and *C. hsingshangensis*.

Sample	IC_50_ [µg/mL]			
	Lipoxygenase Inhibition	Hyaluronidase Inhibition	COX-1 Inhibition	COX-2 Inhibition
CHi	28.73 ± 2.01 ^e,f,g,h,i,l^	15.09 ± 0.61 ^e,g,h,i,k^	24.77 ± 0.35 ^e,f,g,h,i,j^	19.95 ± 0.08 ^e,f,g,h,i,j^
CHi-1	129.46 ± 6.35 ^a,i,l^	24.95 ± 0.52 ^a,e,f,g,h,i,k^	62.54 ± 1.02 ^a,e,f,g,h,i,j^	100.36 ± 1.27 ^a,e,f,g,h,i,j^
CHi-2	19.77 ± 1.56 ^b,i,l^	13.19 ± 0.08 ^a,b,e,f,g,h,i,k^	11.15 ± 0.42 ^a,b,e,f,g,h,i,j^	16.68 ± 0.19 ^a,b,e,f,g,h,i j^
CHi-3	45.69 ± 4.21 ^a,b,c,i,l^	10.28 ± 0.23 ^a,b,c,e,f,g,h,i,k^	46.71 ± 0.69 ^a,b,c,e,f,g,h,i,j^	38.50 ± 0.32 ^a,b,c,e,f,g,h,i,j^
CHi-4	97.12 ± 3.86 ^a,b,c,d,i,l^	19.80 ± 0.51 ^a,b,c,d,e,f,g,h,i,k^	45.35 ± 0.47 ^a,b,c,e,f,g,h,i,j^	81.69 ± 0.78 ^a,b,c,d,e,f,g,h,i,j^
CHs	11.06 ± 1.14 ^l^	6.82 ± 0.15 ^k^	13.02 ± 0.17 ^j^	9.21 ± 0.09 ^j^
CHs-1	72.15 ± 1.57 ^e,l^	14.76 ± 0.19 ^e,k^	34.12 ± 0.28 ^e,j^	57.59 ± 0.65 ^e,j^
CHs-2	4.15 ± 0.31 ^e,f^	1.17 ± 0.02 ^e,f,k^	6.39 ± 0.04 ^e,f,j^	5.09 ± 0.06 ^e,f^
CHs-3	11.56 ± 0.98 ^f,g,l^	7.41 ± 0.05 ^e,f,g,k^	15.56 ± 0.13 ^e,f,g,j^	10.32 ± 0.25 ^e,f,g,j^
CHs-4	5.72 ± 0.26 ^e,f,h^	1.89 ± 0.05 ^e,f,g,h,k^	9.54 ± 0.04 ^e,f,g,j^	15.03 ± 0.19 ^e,f,g,j^
IND	nt	nt	4.34 ± 0.05	3.82 ± 0.09
EGCG	nt	6.25 ± 0.02	nt	nt
NDGA	5.89 ± 0.15	nt	nt	nt

CHi—crude extract of *C. hissaricus*, CHs—crude extract of *C. hsingshangensis*, CHi-1/CHs-1—water fraction, CHi-2/CHs-2—diethyl ether fraction, CHi-3/CHs-3—butanol fraction, CHi-4/CHs-4—ethyl acetate fraction; EGCG—epigallocatechin gallate, NDGA—nordihydroguaiaretic acid; IND—Indomethacin. Statistical analysis: a—significantly different results compared to CHi; b—significantly different results compared to CHi-1; c—significantly different results compared to CHi-2; d—significantly different results compared to CHi-3; e—significantly different results compared to CHs; f—significantly different results compared to CHs-1; g—significantly different results compared to CHs-2; h—significantly different results com-pared to CHs-3; i—significantly different results compared to CHs-4; j—significantly different results compared to IND; k—significantly different results compared to EGCG; l—significantly different results compared to NDGA; One-Way ANOVA test, followed by a Tukey’s multiple comparison test, *p* < 0.05.

**Table 8 cells-11-00367-t008:** Minimum inhibitory concentration (MIC) and minimum bactericidal concentration of the fractions of *C. hsingshangensis* crude extract.

Sample	*S. aureus* ATCC 25923		*S. epidermidis* ATCC 12228		*C. acnes* PCM 2400		*C. acnes* PCM 2334		*C. acnes* ATCC 11827		*C. granulosu* PCM 2462	
	MIC [µg/mL]	$\frac{MBC}{MIC}\;$	MIC [µg/mL]	$\frac{MBC}{MIC}\;$	MIC [µg/mL]	$\frac{MBC}{MIC}\;$	MIC [µg/mL]	$\frac{MBC}{MIC}\;$	MIC [µg/mL]	$\frac{MBC}{MIC}\;$	MIC [µg/mL]	$\frac{MBC}{MIC}$
CHi	>1000	>4	>1000	>4	>1000	>4	>1000	>4	>1000	>4	>1000	>4
CHs	>1000	>4	>1000	>4	>1000	>4	>1000	>4	>1000	>4	>1000	>4
CHs-1	>1000	>4	1000	>4	500	>4	1000	>4	1000	>4	500	>4
CHs-2	500	>4	1000	>4	31.25	8	62.5	8	125	8	62.5	8
CHs-3	>1000	>4	>1000	>4	1000	>4	500	>4	1000	>4	1000	>4
CHs-4	1000	>4	>1000	>4	250	>4	250	>4	125	>8	250	>4

CHs-1—water fraction of *C. hsingshangensis*, CHs-2—diethyl ether fraction of *C. hsingshangensis*, CHs-3—butanol fraction of *C. hsingshangensis*, CHs-4—ethyl acetate fraction of *C. hsingshangensis.*

## Data Availability

Data available on request.

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
