# Peer review of "The Anti-Acne Potential and Chemical Composition of Two Cultivated Cotoneaster Species"

_cells, 2022, doi:10.3390/cells11030367_

Round 1

Reviewer 1 Report

The article “The anti-acne potential and chemical composition of two cultivated Cotoneaster species” by B. KrzemiÅ„ska and colleagues investigates the composition and the antioxidant and anti-inflammatory properties of different solvent extracts from Cotoneaster hsingshangensis and Cotoneaster hissaricus.

The article is interesting and loaded of data but the way it is presented is quite heavy and difficult to read. Furthermore, some improvements in the analysis of the data would make the results even more significative.

For an easier reading, I would report the data as graphs in the main text and include the tables with the numerical details in the supplementary information.

On the other hand, it would be interesting to include e a representative chromatogram for the LC-MS analysis (as done for the GC-MS analysis).

The discussion of the results and its comparison with data already reported in the literature from other authors should also include their method of extraction as it deeply influences the outcome of the extraction (as also discussed by the authors in the preliminary part of this study).

A statistical analysis must be performed (e.g. ANOVA) to reveal the significance of the obtained data (e.g. in the composition and in the in vitro tests).

Also, given the interesting amount of data, the use of multivariate analysis (PCA, PLS-DA, heatmap, correlation analysis, etc.) could yield interesting information about abundance of the components in the different extracts and correlations between the composition of the extracts and the obtained biological data.

Finally, many English errors are present and must be corrected (also the form).

Several editing errors are also present and must be corrected.

Line 17: it is “i.e.” not “i.a.”

Lines 423-429 and elsewhere: the chemical names of the compound do not need to be in capital case (if they are not starting the sentence), while the letters indicating the stereochemistry must be reported in capital letter, i.e. “2-deoxy-D-galactose”, “1,3:2,5-dimethylene-L-rhamnitol”, “phytol acetate”, etc. Please check carefully all the chemical names as many errors are present.

Reviewer 2 Report

Current study by KrzemiÅ„ska, provides a detail characterization of composition of extracts from C. hissaricus and C. hsingshangensis.  Antioxidant, anti-inflammatory, antimicrobial, and  cytotoxic activities of these extracts were evaluated. It is a well-designed study on phenolic compounds from Cotoneaster species and may provide significant importance in pharmacy and medicine. Furthermore, these findings can direct future research in animal models, and may provide insight into novel therapeutic targets of acne vulgaris. It is a well written article. I have few comments:

 Author should include a brief description on C. acnes, a potential factor associated with severity of acne. Author could follow Keshari et al, “Prospects of acne vaccines targeting secreted virulence factors of Cutibacterium acnes”.

Relevant citation should be provided for the paragraph, “Evaluation of parameters associated with oxidative stress such as: nitric oxide (NO), xanthine oxidase (XO), malondialdehyde (MDA), superoxide dismutase (SOD), and catalase (CAT) in the venous blood of patients using spectrophotometrically methods indicates that oxidative damage of tissues is a significant etiological factor of acne.”

In table 1, the total Flavonoid Content of, CHi-4 – ethyl acetate fraction of C. hissaricus was comparable to CHs-3 – butanol fraction of C. hsingshangensis. Author should include the results from Zones of bacterial growth inhibition of the Cotoneaster fractions, such as, CHi-2 – diethyl ether fraction of C. hissaricus, CHi-3 – butanol fraction of C. hissaricus in the table 8.

In Figure 3. Zones of bacterial growth inhibition of the Cotoneaster extracts x axis which sample group 6, 7 A.B,C and D represents. Detailed information should be included.

Propionibacterium acnes (P. acnes) in the manscripot should be replaced with updated nomenclature, Cutibacterium acnes (C. acnes).

In table 9, MIC and MBC value of C. hsingshangensis crude extract should be provided to make an easier comparison with cytotoxicity data in human fibroblasts.

It will be interesting to detect if these extracts show any specificity against Cacnes cell aggregation at certain concentrations and could prevent  Cacnes biofilm formation.

Although data from inhibition of lipoxygenase activity  supports anti-inflammatory activityof these extract’s. There are various targets associated with inflammatory activation by the C. acnes. Detection of  pro-inflammatory cytokine levels such as IL-8 in human keratinocytes will provide additional information on anti-inflammatory activity of these extracts.

Round 2

Reviewer 1 Report

Despite the article still needs some minor formatting and editing work, the authors did a great effort in improving the manuscript that, in my opinion, is now suitable for publication. 

Reviewer 2 Report

Article by KrzemiÅ„ska et al, investigates the composition and the antioxidant and anti-inflammatory properties of different solvent extracts from Cotoneaster hsingshangensis and Cotoneaster hissaricus. It is a well-designed study on phenolic compounds from Cotoneaster species and may provide its significant importance in pharmacy and medicine. Moreover, these findings can direct future research in animal models, and may provide insight into novel therapeutic targets of acne vulgaris. It is a well written article.